# RNA binding protein SYNCRIP maintains proteostasis and self-renewal of hematopoietic stem and progenitor cells

Florisela Herrejon Chavez [1,2,16], Hanzhi Luo[1], Paolo Cifani[1,3], Alli Pine[4], Eren L. Chu[1,5], Suhasini Joshi[6], Ersilia Barin[1,6,7], Alexandra Schurer [1], Mandy Chan[1], Kathryn Chang[1], Grace Y. Q. Han[1], Aspen J. Pierson[1], Michael Xiao[8], Xuejing Yang[1], Lindsey M. Kuehm[9], Yuning Hong[10], Diu T. T. Nguyen[1,11], Gabriela Chiosis [6], Alex Kentsis[1,12,13], Christina Leslie [4], Ly P. Vu [1,14,15,16,17] ✉ & Michael G. Kharas [1] ✉

Tissue homeostasis is maintained after stress by engaging and activating the hematopoietic stem and progenitor compartments in the blood. Hematopoietic stem cells (HSCs) are essential for long-term repopulation after secondary transplantation. Here, using a conditional knockout mouse model, we revealed that the RNA-binding protein SYNCRIP is required for maintenance of blood homeostasis especially after regenerative stress due to defects in HSCs and progenitors. Mechanistically, we find that SYNCRIP loss results in a failure to maintain proteome homeostasis that is essential for HSC maintenance. SYNCRIP depletion results in increased protein synthesis, a dysregulated epichaperome, an accumulation of misfolded proteins and induces endoplasmic reticulum stress. Additionally, we find that SYNCRIP is required for translation of *CDC42 RHO-GTPase*, and loss of SYNCRIP results in defects in polarity, asymmetric segregation, and dilution of unfolded proteins. Forced expression of CDC42 recovers polarity and in vitro replating activities of HSCs. Taken together, we uncovered a post-transcriptional regulatory program that safeguards HSC self-renewal capacity and blood homeostasis.

The life-long self-renewal activity of stem cells is fundamental for maintaining normal and healthy function of almost all organs. Hematopoietic stem cells (HSCs) retain the highest self-renewal potential among all the blood cells. However, mouse and human HSCs have phenotypic and functional heterogeneity, which includes lineage bias, self-renewal potential and differential ability to respond to stress. Recent studies in mouse and human further identified a reserve HSC population, which is endowed with the highest self-renewal potential and is essential for responding to stress[1,2]. These HSCs must sustain themselves from both internal and external insults that include replicative, genotoxic and physiological stress (e.g., inflammation, infections and aging). Continuous exposures to these conditions and failure

to mitigate adverse impact of stress can result in HSC exhaustion, defective blood production, clonal hematopoiesis and outgrowth of pre-malignant clones[3]. While chronic stress and severe perturbations ultimately lead to clearance of damaged HSCs[4,5], in response to transient distresses, HSCs can mount appropriate cellular responses that allow them to continue replenishing the system while preserving self-renewing capacity[6,7].

The adaptive response of HSCs to stress signals is maintained through a specific gene expression program[8–10] and cellular metabolic state[11]. It has been demonstrated that HSCs possess a distinct metabolic profile, exhibiting minimal activity of macromolecule anabolism[12] and protein synthesis[13], which is notably different from downstream

progenitor populations. Importantly, these features render HSCs much more vulnerable to metabolic stress induced by physiological changes. For example, interference in lipid metabolism specifically impacts HSC activities[14]. A moderate increase in protein production in HSCs disrupts protein homeostasis, thereby diminishing HSC self-renewal[15]. On the other hand, HSCs have been characterized to have heightened autophagy and stress response activities[6,7,16]. HSCs employ these protective cellular mechanisms to quickly resolve crisis and restore cellular homeostasis. Thus, ensuring cellular integrity during stress responses is key for maintenance of HSC function.

RNA binding proteins (RBPs) constitute a group of functional proteins with the unique ability to bind directly to mRNA and influence its fate – how it is processed, where it is localized, its half-life and how it is translated[17]. RBPs play a central role in modulating post-transcriptional gene expression regulation, the regulatory layer which provides diversity and responsiveness to the acute perturbations from both internal and external cellular environments. RBPs have emerged as an important class of regulators on stem cell fate decisions[18,19], particularly in the hematopoiesis system[20]. Several RBPs including Lin28[21] and MSI2[22,23] are involved directly in promoting HSCs' self-renewal. Despite the increasing evidence supporting a central role for RBPs in HSC biology, the identities and underpinning mechanisms of RBPs that govern HSCs remains poorly characterized.

To expand our understanding of RBP's function in the hematopoietic system, we had previously performed an RBP-focused in vivo screen and identified SYNCRIP as a critical RBP that controls the leukemic gene expression program in myeloid leukemia[24]. Here, using a murine genetic conditional knockout (cKO) model, we investigated SYNCRIP's role in adult HSCs. We demonstrated that loss of SYNCRIP impaired blood homeostasis and self-renewal of the hematopoietic stem and progenitor compartments, especially during a stress response. Single-cell RNA-seq (scRNA-seq) analysis of hematopoietic stem/progenitor cells (HSPCs) in the transplantation setting revealed a strong induction of unfolded protein responses upon SYNCRIP depletion within the HSC populations. We further showed that SYNCRIP is required to maintain proteostasis in HSCs. We then employed a multi-omic approach to comprehensively define SYNCRIP's targets and functional downstream pathways in HSCs. Overall, we revealed the functional requirement for SYNCRIP in preserving HSC's self-renewal under regenerative stress while uncovered a major role of RBPs in the control of cellular protein homeostasis.

## Results

### SYNCRIP has modest effects in steady-state hematopoiesis

We observed that Syncrip is highly expressed within the hematopoietic stem and progenitor compartments and decreased in more differentiated myeloid and lymphoid populations (Fig. S1A). Thus, to directly investigate the physiological role of SYNCRIP in adult hematopoietic cells, we developed a conditional Syncrip knockout (KO) allele. The targeting vector was designed to place two flox-P sites flanking exon 3 and 4 of genomic Syncrip locus. This allows for generation of a premature stop codon upon Cre-loxP activation (Fig. S1B and Fig. 1A). We crossed Syncrip[f/f] mice to the interferon (IFN) -α-inducible Mx-1-Cre mice to create Syncrip[f/f] Mx-1-Cre + (cKO) and wild type (WT) control Syncrip[f/f] Mx-1-Cre-. We injected 6-8-week-old Syncrip[f/f] Mx-1-Cre + and control Syncrip[f/f] Mx-1-Cre- with two rounds of poly(I:C) (pIpC) to induce excision within the Syncrip alleles. We obtained highly efficient depletion of SYNCRIP as demonstrated by undetectable SYNCRIP protein in bone marrow (BM) cells of KO Syncrip[Δ/Δ] (Fig. 1B).

To assess the effects of Syncrip deletion in adult hematopoiesis, we examined the hematopoietic compartments of WT Syncrip[f/f] and KO Syncrip[Δ/Δ] at multiple time points post pIpC. At 24 weeks post pIpC, loss of SYNCRIP resulted in reduction of total blood count and lymphocyte count but did not significantly impact the count of red blood

cells, hemoglobin and platelets in peripheral blood (Fig. S1C–G). However, we observed no significant change in total cellularity in bone marrow (BM) as well as overall spleen and liver weights of SYNCRIP deficient mice at both short-term (3 weeks) and long term (24 weeks) timepoints post pIpC (Fig. 1C and S1H). Within the HSPC compartment we observed an increase in the LSK frequency, however we noted no significant change in the absolute number of LSK (Lin-Sca+ckit + ) cells (Fig. 1D, E, Fig. S1I). Additionally, we found no significant reduction in frequencies or absolute numbers of HSCs (LSK CD150 + CD48-) and multiple potent progenitor (MPP) populations as well as downstream progenitors after Syncrip deletion (Fig. 1F and Fig. S1I, J). In agreement with our previous observation using a CRISPR-mediated KO of Syncrip murine model[24,25], these data indicate that SYNCRIP contributes quantitatively to lymphocyte count and LSK frequency but, hematopoiesis at steady-state is relatively preserved.

### SYNCRIP is required for long term reconstitution

To examine the function of SYNCRIP in HSPCs, we first performed in vitro colony forming and re-plating assays from WT Syncrip[f/f] and KO Syncrip[Δ/Δ] hematopoietic cells. While we observed only a modest reduction in number of Syncrip[Δ/Δ] colonies in the 1st round of plating (Fig. S1K), deletion of Syncrip resulted in a significant decrease in colony numbers in the 2nd and 3rd re-plating (Fig. S1L). This data suggests that depletion of SYNCRIP diminishes the in vitro self-renewal potential of HSPCs.

To test the functional requirement of SYNCRIP for in vivo reconstitution, we transplanted BM cells from WT Syncrip[f/f] and KO Syncrip[Δ/Δ] into lethally irradiated, congenic CD45.1 recipient mice and followed donor engraftment. In a non-competitive transplantation, we observed a modest reduction of donor chimerism at the early time points (8wks) and a larger reduction of donor chimerism in both PB and BM over longer periods (16 and 24 wks), (Figs. S1M, N, G–I). The partial reduction was also observed at the level of LSK, myeloid progenitors (MP) and phenotypic primitive hematopoietic stem cells (HSC LSK CD150 + CD48-) and multipotent progenitors (MPP1-LSK CD150-CD48-, MPP2-LSK CD150 + CD48 + and MPP4- LSK CD150-CD48 + ) (Fig. 1J). Within the donor compartment, we found no change in frequencies of LSK and HSCs populations (Fig. S1O–P). However, in a competitive transplantation, there was a drastic loss of Syncrip[Δ/Δ] engraftment (Fig. 1K and S1Q–R). Syncrip[Δ/Δ] donor cells were largely out-competed by control cells across all stem and progenitor populations (Fig. 1L). These data indicate that Syncrip[Δ/Δ] HSPCs exhibit reduced repopulating potentials compared to the control cells.

To control for toxicity from Cre expression and activity in addition to control for IFN response due to the pIpC treatment, we evaluated WT Syncrip[f/f] and heterozygous KO Syncrip[f/Δ]. We found no impact on the ability of Syncrip[f/Δ] cells to engraft in a primary non-competitive transplant (Fig. S1S, T). Interestingly, in a competitive transplantation setting, loss of one Syncrip allele showed mild reduction in chimerism, suggesting that there is a dosage-dependent effect of Syncrip loss on fitness of HSPCs (Fig. S1U). Thus, to further evaluate effects of Syncrip deletion on self-renewal capacity of HSCs, we isolated BM cells from WT Syncrip[f/f] and KO Syncrip[Δ/Δ] primary recipients and transplanted into secondary recipients. While Syncrip[f/f] cells maintained ~50% donor chimerism, Syncrip[Δ/Δ] donors completely failed to engraft in the HSPC compartments and in all mature lineages (Fig. 1M and S1V). Taken together, these data strongly indicate that SYNCRIP is critical for maintenance of self-renewal in HSCs.

### SYNCRIP's role during stress hematopoiesis is autonomous

Given the observed requirement for SYNCRIP's function in hematopoietic reconstitution, we further assessed whether the phenotype is cell autonomous. We transplanted 6–8-week-old Syncrip[f/f] Mx-1-Cre + and control Syncrip[f/f] Mx-1-Cre- into lethally irradiated congenic recipients to exclude potential effects of SYNCRIP loss of function in

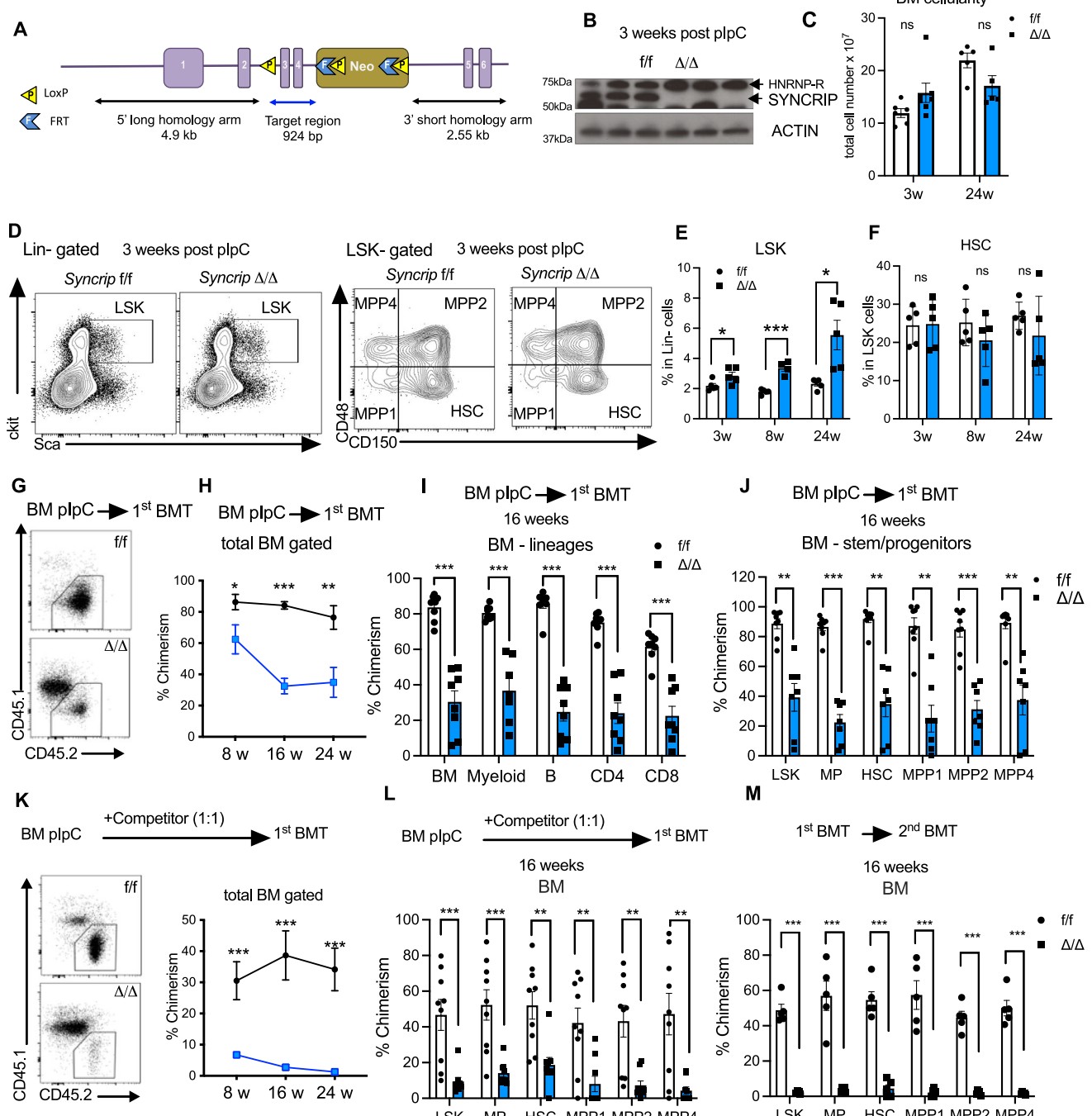

**Fig. 1 | SYNCRIP is dispensable for steady-state hematopoiesis but required for long-term HSC self-renewal. A** Targeting strategy to create *Syncrip* conditional knockout mouse (cKO). **B** Immunoblots showing SYNCRIP KO in bone marrow of *Syncrip*$^{Δ/Δ}$ mice 3 weeks post pIpC injections. ACTIN as loading control. **C** Bone marrow (BM) cellularity *Syncrip*$^{f/f}$ ($n = 6$) and *Syncrip*$^{Δ/Δ}$ ($n = 5$) mice at 3- and 24 weeks post pIpC. **D** Representative flow analysis of stem/progenitor compartments *Syncrip*$^{f/f}$ and *Syncrip*$^{Δ/Δ}$ mice 3 weeks post pIpC. LSK (Lin-Sca+ckit +); HSC – hematopoietic stem cell (LSK CD48-CD150 +); MPP-multipotent-progenitor; MPP1-LSK CD48-CD150-; MPP2-LSK CD48 + CD150 +; MPP4-LSK CD48 + CD150-. **E** Frequencies of LSK cells *Syncrip*$^{f/f}$ and *Syncrip*$^{Δ/Δ}$ mice at 3- ($p = 0.047$), 8-($p = 0.000015$) and 24 ($p = 0.011$), weeks post pIpC ($n = 5$/genotype). **F** Frequencies of HSCs in *Syncrip*$^{f/f}$ and *Syncrip*$^{Δ/Δ}$ mice at 3-, 8- and 24 weeks post pIpC ($n = 5$/ genotype). **G, H** Chimerism of donor-derived BM cells from *Syncrip*$^{f/f}$ or *Syncrip*$^{Δ/Δ}$ ($n = 2$ donor, and $n = 10$ recipients/genotype). **G** Representative flow plots **H** Quantitative analysis at 8- ($p = 0.035$), 16- ($p < 0.000001$) and 24- ($p < 0.0005$) weeks post-transplantation. **I** Donor chimerism in BM mature populations at 16 weeks post-transplantation: total BM ($p = 0.000002$) Myeloid (Mac1 + Gr1 +)

($p = 0.00016$), B (B220 +) ($p < 0.000001$), CD4 (CD4 +) ($p < 0.000001$); CD8 (CD8 +) ($p < 0.000016$) cells ($n = 8$/ genotype). **J** Donor chimerism in stem/progenitor compartments at 16 weeks post-transplantation: LSK ($p = 0.000146$), MP ($p < 0.000001$), MPP1 ($p = 0.000039$), MPP2 ($p = 0.000011$), and MPP4 ($p = 0.000219$) ($n = 8$/ genotype). **K** Donor chimerism in BM cells ($n = 2$ donor, and $n = 10$ recipients/genotype). Left: representative flow plots. Right: Quantitative analysis at 8-($p = 0.000942$), 16-($p = 0.000338$) and 24-($p = 0.000194$) weeks post-transplantation. **L** Donor chimerism in stem/progenitor compartments at 16 weeks post-transplantation. LSK $p = 0.000718$, MP ($p = 0.000498$), HSC ($p = 0.001521$), MPP1 ($p = 0.002307$), MPP2 ($p = 0.001265$), MPP4 ($p = 0.002243$). **M** Donor chimerism in BM of secondary recipient at 16 weeks post-transplantation ($n = 2$ donor, and $n = 10$ recipients/genotype). LSK ($p < 0.000001$), MP ($p = 0.000211$), HSC ($p = 0.00001$), MPP1 ($p = 0.000145$), MPP2 ($p = 0.000002$), MPP4 ($p = 0.000004$). All plots show *Syncrip*$^{f/f}$ as black circles and *Syncrip*$^{Δ/Δ}$ is represented as blue (box or symbol) squares. Source data are provided as Source Data File. All data represent mean±s.e.m. $p$ values were calculated by two-tailed $t$ test unless specified. *$p < 0.05$, **$p < 0.01$, ***$p < 0.001$ and ns: not significant.

the bone marrow environment. We obtained equally high engraftment of both genotypes at 6 weeks post-transplantation and efficient depletion of SYNCRIP in $Syncrip^{\Delta/\Delta}$ BM after pIpC injections (Fig. 2A, B). Similar to what we observed with primary $Syncrip^{\Delta/\Delta}$ BM cells, despite very little effect on number of colonies formation in the 1st plating, ablation of SYNCRIP significantly decreased colony formation in the 2nd and 3rd in vitro re-platings (Fig. S2A, B). At the same time, upon $Syncrip$ deletion, we observed a continuous decrease of donor chimerism in $Syncrip^{\Delta/\Delta}$ recipients over extended timepoints (Fig. 2B and S2C, D). The effects of $Syncrip$ deletion were not lineage specific as chimerism reduction was observed across all mature lineages and HSCPs (Fig. 2C). Except for MPP1, there was no significant decrease in frequencies of HSPC populations within the donor compartments (Fig. 2D). We noted that while deletion of $Syncrip$ did not lead to any overt change in the hematopoietic compartment (Fig. 1C and S1C–J), transplanted $Syncrip^{\Delta/\Delta}$ (Tx-$Syncrip^{\Delta/\Delta}$) exhibited mild but significant defects, suggesting that hematopoietic cells undergone transplantation are more susceptible to the loss of SYNCRIP.

As sustained engraftment was observed in the primary transplanted animals, we further tested the response of $Syncrip^{\Delta/\Delta}$ HSCs to hematopoietic stress. We subjected $Syncrip$ deficient HSCs to repopulation stress by performing a secondary bone marrow transplant (BMT). We observed a dramatic loss in HSC and progenitor's ability to engraft in recipients when $Syncrip$ was deleted (Fig. 2E, F and S2E–G).

To determine whether the reduced chimerism was due to homing deficiency, we treated LSK cells isolated from primary transplanted $Syncrip^{\Delta/\Delta}$ and $Syncrip^{f/f}$ with CFSE prior to secondary transplantation and evaluated the presence of CFSE stained cells within BM of recipients 16 h post-transplantation. We observed a slight trend but no significant reduction in percentage of cells homing to the BM (Fig. S2H, I). At one-week post-secondary transplantation, we observed a modest and significant reduction in engraftment of SYNCRIP depleted cells (Fig. 2G), strongly indicating that the later loss of chimerism is not driven by homing issues but largely by defects in repopulating activity of HSCs and progenitor cells. Next, we examined the response of $Syncrip^{\Delta/\Delta}$ KO mice to myeloablation. $Syncrip^{\Delta/\Delta}$ KO mice were more sensitive to stress induced by lethal irradiation and displayed worse survival compared to control mice (Fig. 2H). Altogether, loss of $Syncrip$ impairs HSC and progenitor cells' function in response to stress, and most potently under secondary repopulating pressure in transplantation and myelosuppression.

To investigate the cellular mechanisms underlying the observed reduced self-renewal potential in SYNCRIP deficient HSC and progenitors, we initially focused on the known link between increased HSPC activation and loss of self-renewal[26–31]. We performed flow cytometry analysis with Hoechst and Pyronin Y staining to examine whether deletion of $Syncrip$ impairs HSPCs' cell cycle progression. While we observed a marked reduction in the $G_0$ quiescent population and an increase of the S/G2/M fraction in MPP1 cells, we found only a slight, but not significant, change in cell cycle status of $Syncrip^{\Delta/\Delta}$ HSCs (Fig. 2I, J). To further probe if SYNCRIP deletion altered the proliferation of HSCs, we injected transplanted KO $Syncrip^{\Delta/\Delta}$ and WT $Syncrip^{f/f}$ mice with BrdU and traced the incorporation of BrdU in cycling cells within the HSPC compartment. We found no significant change in frequencies of BrdU positive cells across HSCs and most MPP populations. We did see a significant but modest increase of BrdU incorporation in MPP2 (Fig. S1J). These data suggest that the early drop in donor engraftment after $Syncrip$ deletion was associated with enhanced cycling in the MPPs. Importantly, this was limited only to MPPs and not in the HSCs.

To directly examine whether SYNCRIP depletion alters division of HSCs, we plated single HSCs from KO $Syncrip^{\Delta/\Delta}$ and WT $Syncrip^{f/f}$ transplanted mice and followed cell division in vitro using the CellRaft AIR® System (details in methods) for 60 h. While there was a slight increase in the mean value of time to first cell division (12.23 h in

$Syncrip^{\Delta/\Delta}$ vs. 10.16 h in $Syncrip^{f/f}$), the cumulative outputs were not different between the two conditions (Fig. 2K and Fig. S2K). The data suggested that the impact of $Syncrip$ loss is minimal on HSC divisions. To further validate the results, we performed CFSE labeling of LSK cells, engrafted stained cells into recipients and then tracked cell divisions in vivo over a period of one week. The highest CFSE signals in cells indicates cells that either did not divide while reduced CFSE signals reflect progressive dilution of CFSE fluorescence in daughter cells following each cell division (Fig. 2L and Fig. S2L). We observed equal frequencies of high, mid and low CFSE LSKs and HSCs in WT and KO conditions (Fig. 2M and S2M), indicating that SYNCRIP deletion did not affect cell proliferation of either low or high-dividing populations of HSCs. Overall, these data indicate that SYNCRIP does not influence HSC cell cycle or proliferation.

## $Syncrip$ deleted HSCs display an activated stress response

To decipher the effect of $Syncrip$ deletion on cellular identities along the hematopoietic hierarchy and to gain an in-depth assessment of the transcriptomic changes in different cell types upon $Syncrip$ loss, we performed single-cell RNA sequencing analysis (scRNA-seq) of sorted LK cells (Lin-cKit+ cells) from transplanted $Syncrip\Delta/\Delta$ vs. $Syncrip$ f/f mice (described in Fig. 2A). We were able to perform analysis on total of nearly 50,000 cells (Supplementary Data 1). Uniform Manifold Approximation and Projection (UMAP) (Fig. 3A) and t-Distributed Stochastic Neighbor Embedding (t-SNE) (Fig. S3A) analysis identified a majority of previously characterized stem and progenitor clusters including HSC, MPPs and erythroid, myeloid and lymphoid progenitors (Fig. 3A, B-Supplementary Data 2)[32–34]. We observed only modest changes in cellular frequencies of later progenitors i.e., Ba, Mk, Mo1 and Mo2, and no significant change in the frequencies of HSC and MPPs (Fig. 3F and S3C, D-Supplementary Data 4). This data was in line with the phenotypic flow cytometry analysis of different HSPC populations, indicating that there is no major change in HSC cell fates and lineage choices.

We identified two HSC-like populations with both t-SNE and UMAP, which we annotated as an HSC cluster 1 (HSC-C1) and HSC cluster 2 (HSC-C2) (Fig. 3A, S3A, B). Based on gene expression, these population's transcriptome are enriched for signatures associated with HSCs compared to MPP1 and MPP2 (Fig. 3B and Fig. S3E). Trajectory analysis[35] indicated that HSC-C1 is connected mainly with HSC-C2 and with a small portion of early MPP1 (Fig. 3C). In a transplant setting, HSCs can be functionally specified into a functional long-term repopulating low-output or reserve HSCs and differentiating high-output HSCs[1]. Using functional in vivo bar-coding strategies, low-output HSCs have recently been characterized to contain long-term self-renewing potency and are responsible for propagating the hematopoietic system specifically in a secondary transplantation. We performed GSEA analysis to compare the transcriptomic profile of HSC-C1 (HSC-C1 vs. HSC-C2 – Supplementary Data 2 and 3) against transcriptional signatures of low-input and high-input HSCs. Interestingly, HSC-C1 transcriptionally corresponded to the low-output HSC population and were distinct from high-output HSCs (Fig. 3D). Additional GSEA analysis revealed that transcriptome of HSC-C1 closely resembled those of more primitive HSCs (Fig. S3F), HSCs characterized to have serial transplant potential (Fig. S3G) (Rodriguez-Fraticelli et al., 2020), self-renewal HCSs (Fig. S3H) (Pietras et al., 2015), dormant HSCs (Fig. S3I) (Cabezas-Wallscheid, N. et al., 2017) and possess the StemScore signature (Fig. S3J) (Giladi, A. et al., 2018). These results support the previous observation of different sub-HSC populations in the transplant setting where HSC-C1 transcriptionally resembles the HSC cluster characterized with high self-renewal potential. It is however noted that these analyses are based only on transcriptomic profiling and are not functionally defined.

We then examined the specific genes associated with the HSC clusters in the SYNCRIP depleted cells and found an upregulation of

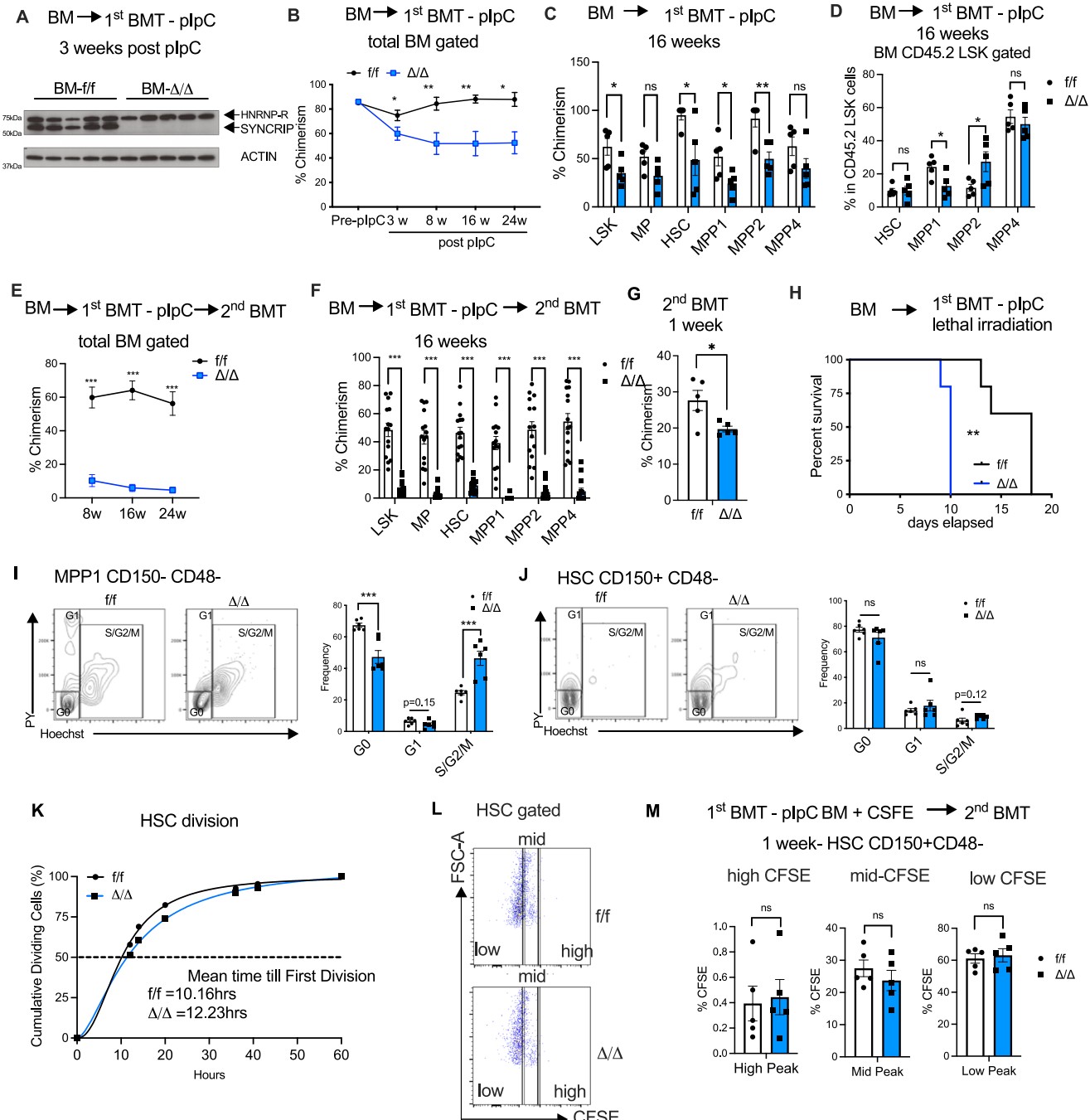

**Fig. 2 | SYNCRIP plays a critical role in stress hematopoiesis. A** Immunoblots showing SYNCRIP KO in engrafted *Syncrip* ^Δ/Δ^ BM cells 3 weeks post pIpC injections. ACTIN as loading control. **B** Chimerism of donor-derived BM cells in recipient mice described in (**A**) at: pre-pIpC, 3-(*p* = 0.037), 8-(*p* = 0.0059), 16-(*p* = 0.0029) and 24-(*p* = 0.0041) weeks post pIpC (*n* = 3 donor, and *n* = 15 recipients/genotype at pre- and 3 weeks; *n* = 9 at 8-, 16- and 24- weeks). **C** Chimerism of donor-derived cells in stem/progenitor compartments of recipient mice described in (**A**) at 16 weeks post pIpC. LSK (*p* = 0.030), MP (*p* = 0.051), HSC (*p* = 0.0254), MPP1 (*p* = 0.0333), MPP2 (*p* = 0.0058), MPP4 (*p* = 0.141). **D** Quantitative summary of frequencies of HSC and MPPs cells within CD45.2+ LSK populations of recipient mice described in (**A**) at 16 weeks post pIpC. HSC (*p* = 0.95), MPP1 (*p* = 0.024), MPP2 (*p* = 0.037), MPP4 (*p* = 0.46). **E** Chimerism of donor-derived BM cells in secondary recipients at 8-, 16- and 24 weeks post-transplantation. (*n* = 3 donor, and *n* = 15 recipients/genotype). All *p* < 0.000001. **F** Chimerism of donor-derived cells in stem/progenitor compartments of secondary recipient mice (described in **E**) at 16 weeks post-

transplantation (*n* = 3 donor, and *n* = 15 recipients/genotype). All *p*-values< 0.000001. **G** Chimerism of donor-derived cells in total BM cells of secondary recipient mice at 1 week post-transplantation (*n* = 7 donor, *n* = 5 recipient/ genotype). *p*-value=0.0256. **H** Kaplan–Meier analysis of survival of WT *Syncrip*^f/f^ and KO *Syncrip*^Δ/Δ^ mice following lethal irradiation (*n* = 5 each). *p*-value=0.0035. **I, J** Cell cycle analysis of (**I**) MPP1 (G0 *p* = 0.00089, G1 *p* = 0.15, S/G2/M *p* = 0.00095) and (**J**) HSCs *Syncrip*^f/f^ and *Syncrip*^Δ/Δ^ mice (*n* = 6 each genotype). Left: Representative flow plots. Right: Quantitative analysis. **K** Cumulative graphs tracking in vitro division of HSCs over the course of 60 h. **L** Representative flow plots showing gating strategy for CFSE stained HSC in recipient mice (described in G) at 1 week post-transplantation. **M** Quantitative summary of data shown in (**L**). Source data are provided as Source Data File. All data represent mean ± s.e.m. *p* values were calculated by two-tailed *t* test unless specified. \**p* < 0.05, \*\**p* < 0.01, \*\*\**p* < 0.001 and ns not significant.

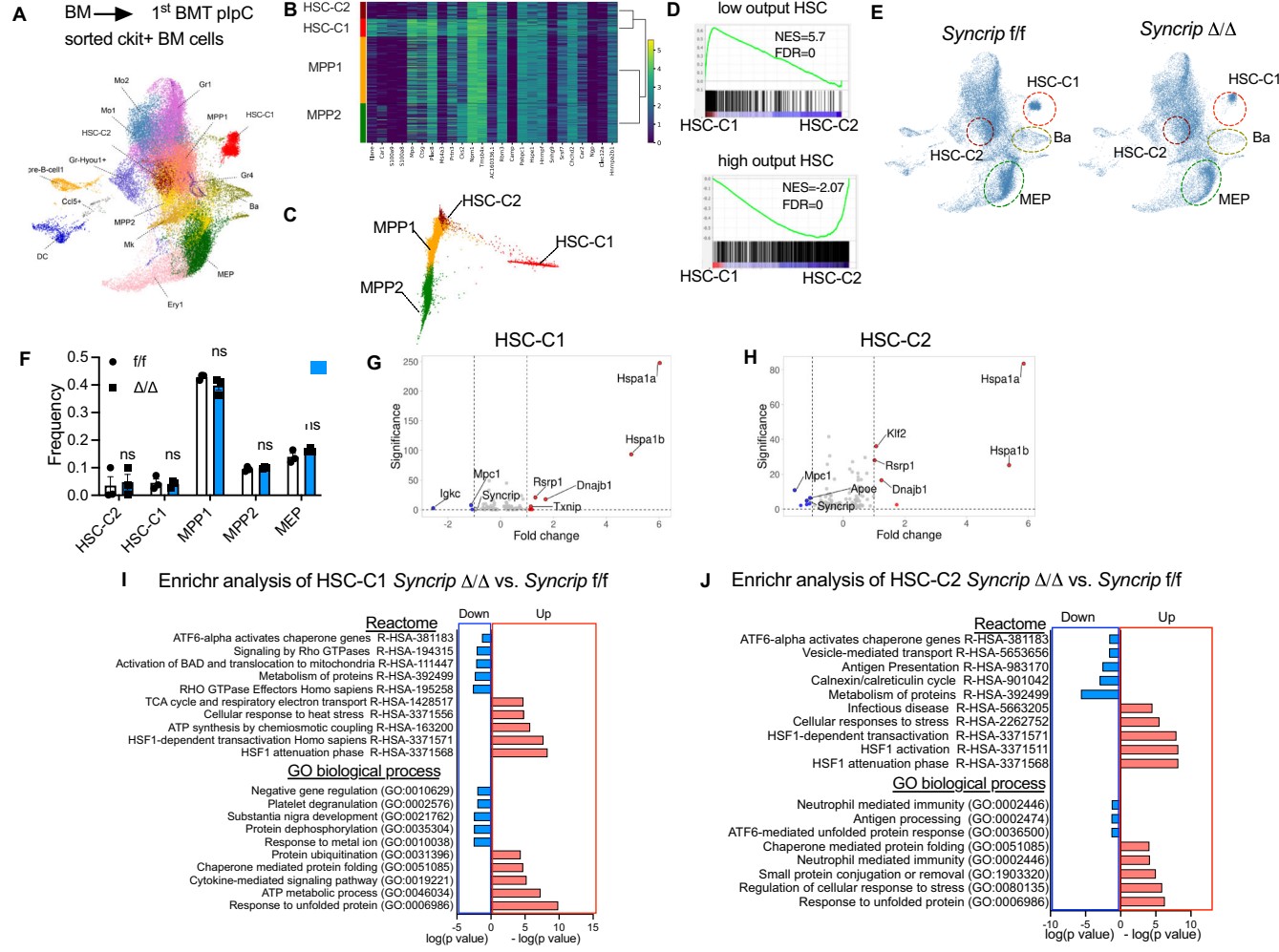

**Fig. 3 | Single-cell RNA sequencing (scRNA-seq) uncovered an activated unfolded protein response in *Syncrip* deficient HSC populations. A** Identification of hematopoietic cell populations within WT *Syncrip*[f/f] (*n* = 3) and KO *Syncrip*[Δ/Δ] (*n* = 3) Lin-ckit+ cells based on UMAP analysis of Single-cell RNA sequencing (scRNA-seq). **B** Gene expression heat map of the highly expressed genes in hematopoietic stem cell-cluster 1 (HSC-C1), hematopoietic stem cell-cluster 1 (HSC-C2), multipotent-progenitor 1 (MPP1) and 2 (MPP2) populations. **C** Reconstruction of the lineage branching among of four early hematopoietic stem/progenitor cells HSC-C1, HSC-C2, MPP1 and MPP2 populations using diffusion pseudotime (DPT) analysis. **D** GSEA analysis of genes differentially expressed between HSC-C1 vs. HSC-C2 in WT *Syncrip*[f/f] mice for enrichment of gene signatures specific for low output and high output HSC. **E** UMAP displays of all hematopoietic clusters of WT *Syncrip*[f/f] and KO *Syncrip*[Δ/Δ] scRNA-seq as described in (**A**). Cluster HSC-C1 shows the most shift. HSC, Ba and MEP clusters were highlighted for comparison. **F** Quantitative summary of

frequencies of different populations defined by scRNA-seq analysis in WT *Syncrip*[f/f] (*n* = 3) and KO *Syncrip*[Δ/Δ] (*n* = 3). Data represent mean ± s.e.m. *p* values were calculated by two-tailed *t* test unless specified. ns not significant. **G, H** Volcano plots showing genes differentially expressed between *Syncrip*[f/f] vs. *Syncrip*[Δ/Δ] within HSC-C1 and HSC-C2 clusters. The most differentially expressed genes are highlighted. **I** Enrichr analysis for GO biological processes and Reactome enrichment of significant (FDR < 0.05) downregulated and upregulated genes within the HSC-C1 population of *Syncrip*[Δ/Δ] vs. *Syncrip*[f/f]. X-axis: -log$_{10}$(p value). Enrichment of down-regulated targets was depicted as negative log$_{10}$(p) and enrichment of upregulated targets was depicted as positive log$_{10}$(p). *p*-values were calculated by Fisher's exact test. **J** Enrichr analysis for GO biological processes and Reactome enrichment of significant (FDR < 0.05) downregulated and upregulated genes within the HSC-C2 population of *Syncrip*[Δ/Δ] vs. *Syncrip*[f/f]. X-axis: -log$_{10}$(*p* value). *p*-values were calculated by Fisher's exact test. Source data are provided as Source Data File.

stress-inducible chaperone genes e.g. *Hspa1a (Hsp70-1)*, *Hspa1b (Hsp70-2)* and *Dnajb1* (Fig. 3G, H and Supplementary Data 5). Pathway enrichment analysis by the Enrichr program (http://amp.pharm.mssm.edu/Enrichr/)[36] indicated a strong activation of cellular response to stress and unfolded proteins, specifically the HSF1-dependent pathways, upon SYNCRIP depletion in HSC-C1 and HSC-C2 populations (Fig. 3I, J and Supplementary Data 6-7). The activation was not seen in MPP1 cells (Fig. S3K), indicating that the phenomenon could be specific for HSCs. Additionally, we observed a downregulation of genes activated by ATF6, a branch of unfolded protein response (UPR) and endoplasmic reticulum (ER) stress response[37], suggesting that the transcriptional activation is specific for HSF1-mediated proteotoxic stress response. We also noted a strong suppression of RHO GTPase effectors and signaling in *Syncrip* KO HSCs. The data suggests that loss

of *Syncrip* resulted in unfolded protein stress in HSCs, thereby contributing to the overall loss of HSC self-renewal and making them vulnerable to repopulation pressure[15].

## SYNCRIP controls protein homeostasis in HSCs

Given the strong induction of transcriptional programs in response to unfolded protein stress in *Syncrip* deficient HSCs, we directly characterized the proteostatic state of HSCs upon SYNCRIP depletion. To measure unfolded proteins in cells, we stained hematopoietic cells with tetraphenylethene maleimide (TMI) and performed flow analysis to assess the accumulation of these proteins[15]. TMI is a cell-permeable dye, which can fluoresce upon binding to free thiol side chains[38]. These thiols within the non-disulphide bonded cysteines normally are not exposed in folded globular proteins. The abundance of accessible

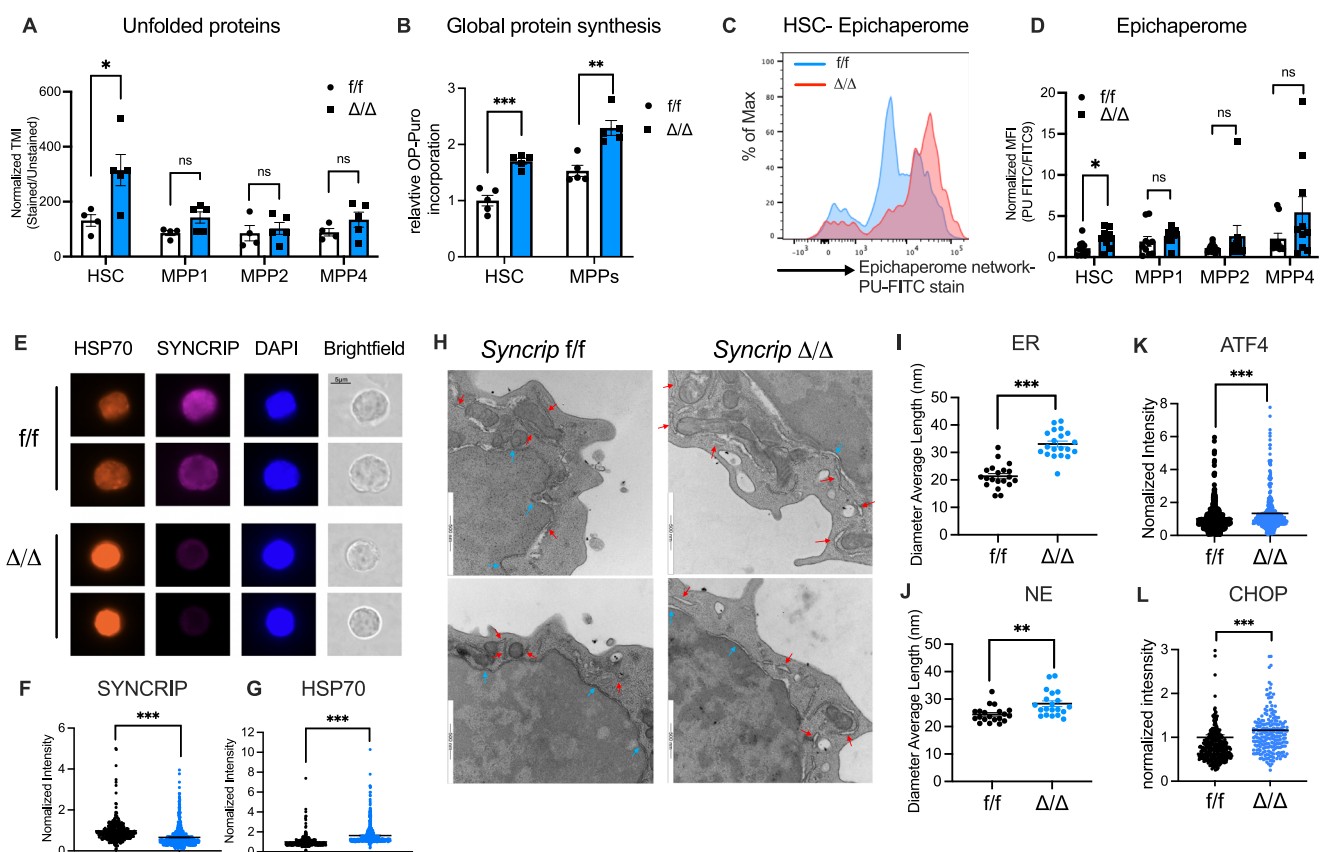

**Fig. 4 | Deletion of SYNCRIP deregulates the proteostasis network in HSCs.**
**A** Quantitative summary of relative tetraphenylethene maleimide (TMI) fluorescent signals in HSC, MPP1, MPP2 and MPP4. TMI signals were used to quantify the unfolded proteins in single hematopoietic cells in *Syncrip*[f/f] (*n* = 4) vs. *Syncrip*[Δ/Δ] (*n* = 5). **B** OP-Puro incorporation in HSC (*p* = 0.00017) and MPP (*p* = 0.0017) populations isolated from *Syncrip*[f/f] (*n* = 5) vs. *Syncrip*[Δ/Δ] (*n* = 5) 1st transplant recipient mice. OP-Puro incorporation was used to quantify the level of global protein synthesis in single hematopoietic cells. **C** Representative histograms of PU-FITC flow analysis of HSCs from *Syncrip*[f/f] vs. *Syncrip*[Δ/Δ]. **D** Quantitative summary of PU-FITC fluorescent signals normalized to FITC control in HSC *p* = 0.0123, MPP1 *p* = 0.32, MPP2 *p* = 0.27 and MPP4 *p* = 0.12. PU-FITC signals were used to quantify the epichaperome signal of hematopoietic cells in *Syncrip*[f/f] (*n* = 10) vs. *Syncrip*[Δ/Δ] (*n* = 10). **E** Immunofluorescence (IF) staining of HSP70 and SYNCRIP in *Syncrip*[f/f] and *Syncrip*[Δ/Δ] HSCs (shown in brightfield). Scale bar 5 µm. **F, G** Quantitative summary of

normalized immunofluorescence reflecting SYNCRIP (*p* = 0.0041) and HSP70 (*p* < 0.0001) protein expression in *Syncrip*[f/f] (*n* = 324) and *Syncrip*[Δ/Δ] (*n* = 654) HSCs (*n* = 5 each genotype). **H** Transmission electron microscopy (TEM) images of HSCs isolated from *Syncrip*[f/f] vs. *Syncrip*[Δ/Δ] mice. Red arrow: endoplasmic reticulum (ER); Blue arrow: nuclear envelope (NE). **I, J** Quantitative summary of diameter average length (nm) of ER (*p* < 0.0001) (**I**) and NE (*p* = 0.0027) (**J**) (*n* = 7 each genotype; total cells *Syncrip*[f/f] *n* = 20 and *Syncrip*[Δ/Δ] *n* = 20). **K, L** Quantitative summary of normalized immunofluorescence reflecting ATF4 and CHOP protein expression in *Syncrip*[f/f] and *Syncrip*[Δ/Δ] HSCs (*n* = 5 each genotype; total cells *Syncrip*[f/f] *n* = 548 and *Syncrip*[Δ/Δ] *n* = 477; *Syncrip*[f/f] *n* = 342 and *Syncrip*[Δ/Δ] *n* = 266) All data *p* < 0.0001. Source data are provided as Source Data File. All data represent mean ± s.e.m. *p* values were calculated by two-tailed *t*-test unless specified. **p* < 0.05, ***p* < 0.01, ****p* < 0.001 and ns: not significant.

thiols thus correlates with the state of unfolded proteome. We observed a significant increase in TMI signals specifically in transplanted HSC, but not MPPs upon *Syncrip* deletion (Fig. 4A and S4A, B), indicating that SYNCRIP is required to maintain high protein quality in HSCs under conditions of regenerative stress.

Given the previously characterized function of SYNCRIP in global translation in leukemia[24], we examined the impact of SYNCRIP depletion in protein synthesis in HSCs and MPPs by measuring the incorporation of OP-puro – a cell permeable analog of puromycin, into nascent polypeptide chains. We observed a higher protein synthesis rate in MPPs vs. HSCs (Fig. S4C). In both *Syncrip* Δ/Δ primary and transplanted mice, *Syncrip* KO MPPs showed a significant increase in protein synthesis in comparison to HSCs. However, only under repopulating pressure in transplantation, *Syncrip* deficient HSCs demonstrated elevated global protein synthesis (Fig. 4B and S4D). These data suggest that SYNCRIP maintains normal protein synthesis activity for HSCs.

To understand the impact of activation of HSF1 pathways and the increase in protein synthesis, we examined the network that facilitates

proper protein folding and homeostasis. HSP90 is a molecular chaperone that interacts with other co-chaperones, adaptors and protein complexes that regulate protein maturation described as the epichaperome network[39]. PU-H71, a chaperone HSP90 inhibitor, has been demonstrated to bind selectively with the epichaperome network. Importantly, increased binding of PU-H71 is observed in aberrant cellular stress response states including in diseases and in myeloid malignancies[39] (Fig. S4E). We found that SYNCRIP depletion elevated the epichaperome network significantly in transplanted HSCs and an increase trend was also observed in MPPs and progenitor cells (Fig. 4C, D and S4E, F). Additionally, we found a significant increase in HSP70 abundance in HSCs (Fig. 4E–G). Furthermore, as a dysregulated epichaperome network is more sensitive to HSP90 inhibition, PU-H71 treatment further reduced colony replating activity of SYNCRIP depleted HSPCs (Fig. S4G). These data suggest that *Syncrip* loss results in activation and dysregulation of the normal protein folding program (HSP90/70).

Increased protein synthesis and accumulation of misfolded proteins can lead to induction of an ER stress response. ER stress can be

directly visualized through transmission electron microscopy (TEM) analysis of sorted HSCs (Fig. 4H). We found that loss of *Syncrip* resulted in an enlarged ER (Fig. 4I) and nuclear envelope (NE) (Fig. 4J), indicating the induction of mild ER-stress in *Syncrip* Δ/Δ HSCs. We also observed an increased expression of downstream effectors of the UPR stress response i.e., ATF4 (Fig. 4K) and CHOP (Fig. 4L), further confirming activation of an ER stress response. Taken together, while we observed increased protein synthesis in both HSCs and MPPs, a dysregulated epichaperome, accumulation of misfolded proteins and increased ER stress was mainly found in the HSC compartment.

## SYNCRIP mRNA targets map to functional pathways in HSCs

We then sought to understand how SYNCRIP directly controls the translational program and alters protein homeostasis in HSCs. To identify direct mRNA targets of SYNCRIP in HSCs and MPPs, we utilized a strategy called hyper-TRIBE[40] that allows us to identify RBP targets using a low-input material. More specifically, we fused SYNCRIP with the enzymatic domain of ADAR to enable (A-I) editing within specific mRNA regions where SYNCRIP is recruited to. Editing events can be identified by variant calling in RNA-sequencing data. We expressed the fused protein SYNCRIP-ADAR (S-ADAR) and control vector (EV) in sorted HSCs (LSK CD150 + CD48-) and MPPs (non-HSCs-LSK) and isolated transduced cells based on GFP positivity at 48 h post-transduction. GSEA analysis for the transcriptional profiles of control EV-transduced HSCs vs. MPPs confirmed the identities of sorted populations (Fig. S5A-Supplementary Data 8). In both HSCs and MPPs, expression of S-ADAR significantly increased the editing frequencies and number of editing events (Fig. S5B-Supplementary Data 9). We applied a strict standard of FDR < 0.05 and differential editing frequency (obtained by subtracting the mean edit frequency of S-ADAR and EV from the mean edit frequency of S-ADAR vs. EV) > 0.1 to call a S-ADAR mediated editing event.

There are 1196 total edit events mapped to 796 genes in HSCs and 831 total edit events mapped to 605 genes in MPPs (Supplementary Data 10). The majority of editing was detected in 3'-UTR compared to the coding region (CDS) of target transcripts (Fig. 5A, B). We mostly found one to two edit sites per transcript and few exceptions of transcripts with more than 4 edited sites (Fig. S5C). There are 534 shared targets of SYNCRIP in both MPPs and HSCs (Supplementary Data 10). We plotted the number of editing events against the expression levels of the transcripts in S-ADAR vs. EV transduced HSC and MPP cells and found no significant correlation (Fig. S5D). Additionally, there was small and no overlap between SYNCRIP targets and genes whose expression is upregulated or downregulated in S-ADAR HSCs and MPPs respectively (Fig. S5E, F). There is also no correlation between number of editing events and expression of edited targets uniquely in MPPs or HSCs (Fig. S5G). These data strongly demonstrate that preference for SYNCRIP binding is independent of mRNA abundancy, supporting the specificity of our hyper-TRIBE approach to identify SYNCRIP direct mRNA targets.

SYNCRIP has been shown to recognize several RNA sequences including polyA[41], UACU splicing element[42] and GGCU/A sequence in miRNAs[43]. However, no consensus SYNCRIP binding motif has been identified due to the lack of a genome-wide assessment for SYNCRIP targets. With the global dataset of SYNCRIP hyper-TRIBE, we performed our customized de novo HOMER analysis[40] to identify SYNCRIP unique binding motifs. Across all targets, we found ACUUAG and UAGG as the most highly enriched motifs (Fig. 5D). We also observed preferential binding sequences at the 3'-UTR and CDS (Fig. S5H-I). All identified motifs were mapped within 250 bp of edit sites, further supporting the specificity and authenticity of the discovered sequences (Fig. 5E and S5J). Within the 3'-UTR, SYNCRIP binds regions containing UAGGU and A-U rich. At CDS, SYNCRIP primarily recognizes G/AGUAAG motif, one of the three canonical U1 sites i.e. GGUAAG, GGUGAG, GUGAGU located adjacent to 5' splice sites[44]. The motifs in

3'UTR are among consensus binding sites for RBP MSI2, which we previously found to cooperatively work with SYNCRIP to drive translation of shared target genes in leukemia cells[24]. These binding patterns indicate that SYNCRIP binds to a range of sequences, which in turn may influence its function in mRNA processing and translation.

We next investigated the relevant molecular and cellular programs regulated by SYNCRIP in HSCs. First, we performed enrichment analysis using Enrichr for the shared SYNCRIP targets in HSCs and MPPs to gain additional insights into pathways associated with SYNCRIP (Fig. 5F – Supplementary Data 11). Significant portions of SYNCRIP targets are involved in mRNA processing and translation with RNA binding activities as well as ubiquitin-mediated proteolysis. Interestingly, we also observed a strong association with RHO GTPase effectors and actin cytoskeleton organization.

To determine whether binding of SYNCRIP influences the abundancy of its targets, we profiled transcriptomic changes in *Syncrip* deficient HSCs by performing RNA-seq analysis of HSCs from *Syncrip*^Δ/Δ vs. *Syncrip*^f/f recipient mice (Fig. S5K and Supplementary Data 12). We functionally annotated the transcriptomic data by performing Gene set enrichment analysis (GSEA)[24,25] using the ranked list of differentially expressed genes in *Syncrip*^Δ/Δ HSCs (Δ/Δ vs. f/f; Supplementary Data 13). We found that gene expression programs associated with differentiation of HSCs toward progenitors or myeloid development[45,46] were suppressed in *Syncrip* deficient HSC while significantly enrichment of genes upregulated in SYNCRIP-depleted HSCs was observed with the set of genes highly expressed in primitive HSCs[46], (Giladi et al., 2018), (Cabezas-Wallscheid, N. et al., 2017) (Fig. S5L-O). In contrast to the reduced self-renewal phenotype previously observed, *Syncrip* deficient HSC displays a transcription profile reflecting a strong preservation of HSCs' stemness and fitness, suggesting a potential transcriptional compensation for these cellular programs. Both gene ontology (GO biological process) and Reactome analysis of *Syncrip*^Δ/Δ deficient HSCs demonstrated a decrease in expression of genes associated with neutrophil-mediated immunity and regulation of cell cycle at the mitotic while there was an increase in expression of genes involved in respiratory electron transport, ribosome biogenesis and translation (Fig. 5G-Supplementary Data 14). Similar to what we observed in SYNCRIP-ADAR overexpression (Fig. S5E, F), there was little overlap between SYNCRIP direct targets and transcripts whose expression is impacted by loss of *Syncrip* (Fig. 5H and S5P). These data indicate that SYNCRIP does not exhibit a strong transcriptional control and potentially plays a more dominant role in post-transcriptional regulation.

To determine the impact of *Syncrip* loss on the proteome of HSPCs, we sorted LSKs from *Syncrip*^Δ/Δ vs *Syncrip*^f/f recipient mice and used label-free quantitative high-resolution mass spectrometry proteomics for differential proteome analysis. We quantified more than 2000 proteins per condition with at least 3 unique peptides per protein (FDR < 0.01, Supplementary Data 15). Interestingly, in contrast to the transcriptomic profile (Fig. S5L-O), GSEA analysis of differentially expressed proteins in *Syncrip*^Δ/Δ vs. *Syncrip*^f/f revealed a reduced abundance of proteins associated with HSC features and with cellular response to stress (Fig. 5I, J), reflecting the phenotypes of *Syncrip* deficient HSPCs. We overlapped the lists of SYNCRIP direct binding targets, SYNCRIP effector mRNAs (RNA-seq 2FC; FDR < 0.05) and proteins (Mass spectrometry analysis 2FC, FDR < 0.1) (Supplementary Data 16). We identified 12 and 51 direct targets of SYNCRIP whose both mRNAs and proteins were reduced and whose protein abundancy were impacted by SYNCRIP KO respectively (Fig. 5K, L). ENRICHR analysis of these SYNCRIP targets showed a strong enrichment for RHO GTPase effectors and pathways involved in control of cytoskeleton in addition to RNA binding proteins including MSI2 (Fig. 5M- Supplementary Data 17). We also noticed several proteins associated with GTPase signaling and cell polarity (highlighted in Fig. 5L) including CDC42, RHOA, and RAN to be reduced in *Syncrip* deficient HSPCs. These data

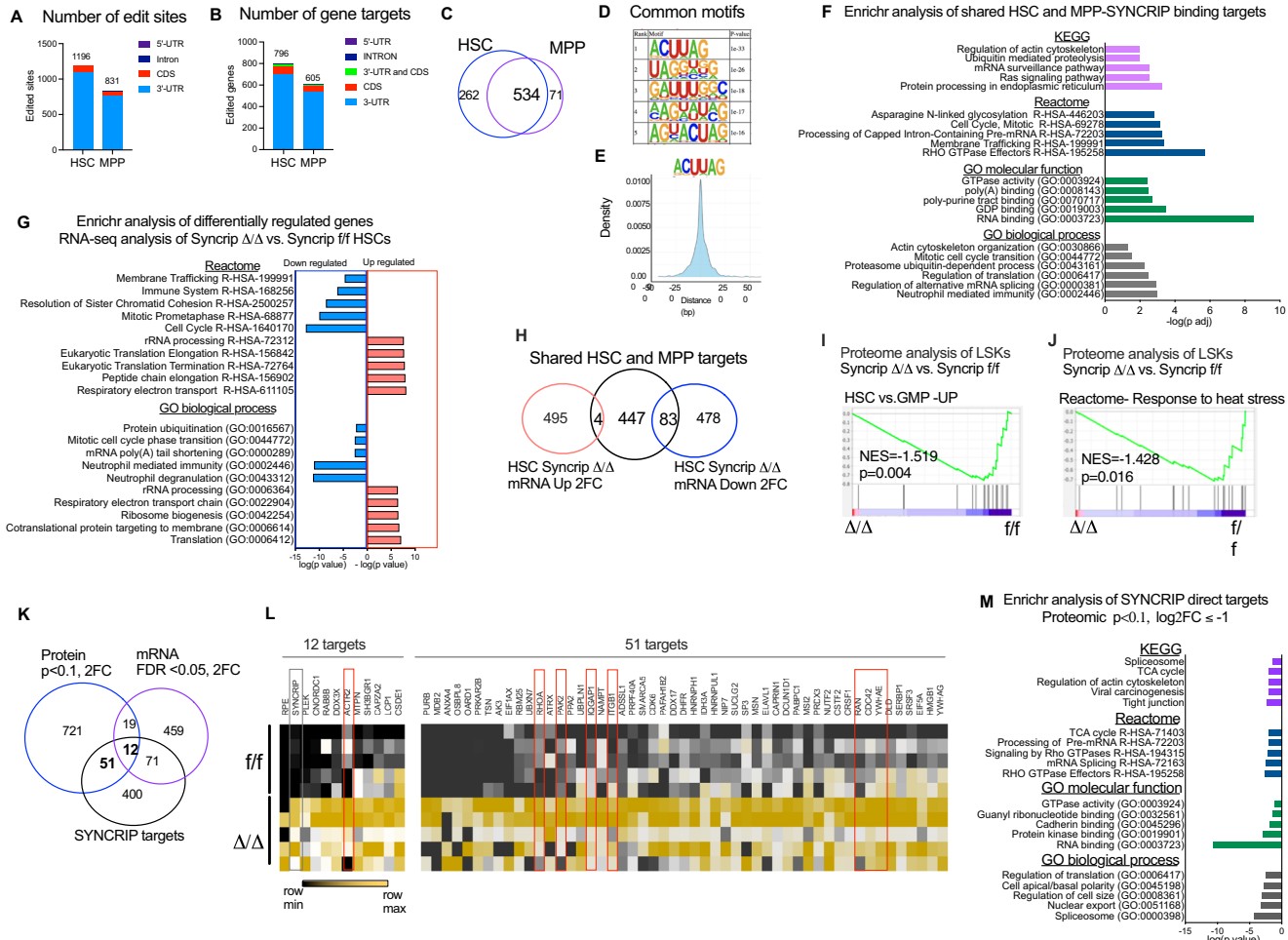

**Fig. 5 | Multi-omics analysis identifies SYNCRIP functional targets in HSCs.**
**A** Number of SYNCRIP-HyperTRIBE significant edit sites (FDR < 0.05 and differential editing frequency (SYNCRIP-ADAR vs. control) ≥ 0.1) and their genic locations in HSC and MPP cells. **B** Number of target genes with edited sites in (**A**) in hematopoietic stem cell (HSC) and multipotent progenitor (MPP). **C** Venn diagram of SYNCRIP target mRNAs identified in HSC and MPP: 534 shared targets, 262 HSC unique targets and 71 MPP unique targets. **D** De novo motif search identifies SYNCRIP-specific binding motifs enriched in mRNA targets with edited sites in both coding regions (CDS and 3'-UTR). **E** Probability density function (PDF) plots showing the distance from edits sites to the nearest SYNCRIP motifs as depicted. **F** Enrichr analysis for GO biological processes, GO molecular function, Reactome and KEGG enrichment of HSC and MPP shared SYNCRIP target genes. X-axis: -log₁₀(p value). **G** Enrichr analysis for GO biological processes and Reactome enrichment of significant (FDR < 0.05) downregulated and upregulated genes in *Syncrip* deficient HSCs. X-axis: Enrichment of downregulated targets negative

$\log_{10}(p)$ and enrichment of upregulated targets positive $\log_{10}(p)$. **H** Venn diagram of SYNCRIP target mRNAs with genes significantly (FDR < 0.05, FC≥2) upregulated (499 genes) and downregulated (n = 561) in *Syncrip* deficient HSCs. **I, J** GSEA analysis showing negative enrichment of proteins significantly downregulated for gene expression signatures upregulated in HSC vs. MPP, Cellular response to heat stress. NES-normalized enrichment score. **K** Venn diagram of SYNCRIP direct target mRNAs (n = 534) with mRNAs significantly downregulated ((FDR < 0.05, FC≤−2: n = 561) in *Syncrip* deficient HSCs and genes of which protein levels are significantly downregulated (FDR < 0.1, FC≤−2: n = 803). **L** Heat map depicting relative protein levels of 12 and 51 overlapping genes highlighted in (**K**) in *Syncrip*^Δ/Δ vs. *Syncrip*^f/f LSKs. Red boxes highlight GTPase/cytoskeleton associated targets. **M** Enrichr analysis for GO biological processes, molecular function, Reactome and KEGG enrichment of SYNCRIP target genes whose proteins are downregulated in *Syncrip* deficient LSKs. X-axis: log₁₀(p value). Source data are provided as Source Data File. For all enricher analysis, p-values were calculated by Fisher's exact test.

point to a direct link between SYNCRIP binding targets and cellular polarity and cytoskeleton organization via control of the RHO GTPase activities.

## SYNCRIP controls cellular polarity via CDC42
GTPase protein family has been implicated in regulation of many aspects of HSPCs function such as proliferation, cytoskeleton organization, polarity, adhesion and migration[47]. Among them, CDC42 is characterized as a central regulator of cytoskeleton dynamics and cell polarity of which proper activities is important to maintain HSC self-renewal and quiescence[48,49]. Given that CDC42 was identified as a direct mRNA target bound by SYNCRIP[50], and protein expression is reduced upon SYNCRIP KO, we further investigated the control of CDC42 expression by SYNCRIP in HSCs. Using immunofluorescence (IF) imaging, we confirmed that depletion of SYNCRIP resulted in a

significant reduction in CDC42 expression in sorted HSCs (Fig. 6A–C). Note that the SYNCRIP antibody also recognizes HNRNP-R as shown in immunoblots (Fig. 1B and supplemental Fig. 1S), resulting in background signal in our IF assays. As previously reported with CDC42 depleted HSCs, we also observed a concurrent reduction in abundance of TUBULIN and a loss of cellular polarity demonstrated by decreased formation of polarized TUBULIN in SYNCRIP deficient HSCs (Fig. 6D, E). The results indicated a direct involvement of SYNCRIP in establishing cellular polarity via its control of CDC42 abundance.

Establishment of cellular polarity is connected with inheritance of fate determinants in daughter cells during division, including during asymmetric versus symmetric cell division[51]. We examined whether aberrant polarization induced by SYNCRIP deficiency alters this process, thus influencing cell fate determination. As we observed that SYNCRIP depletion decreased NUMB protein abundance (Fig. 6F),

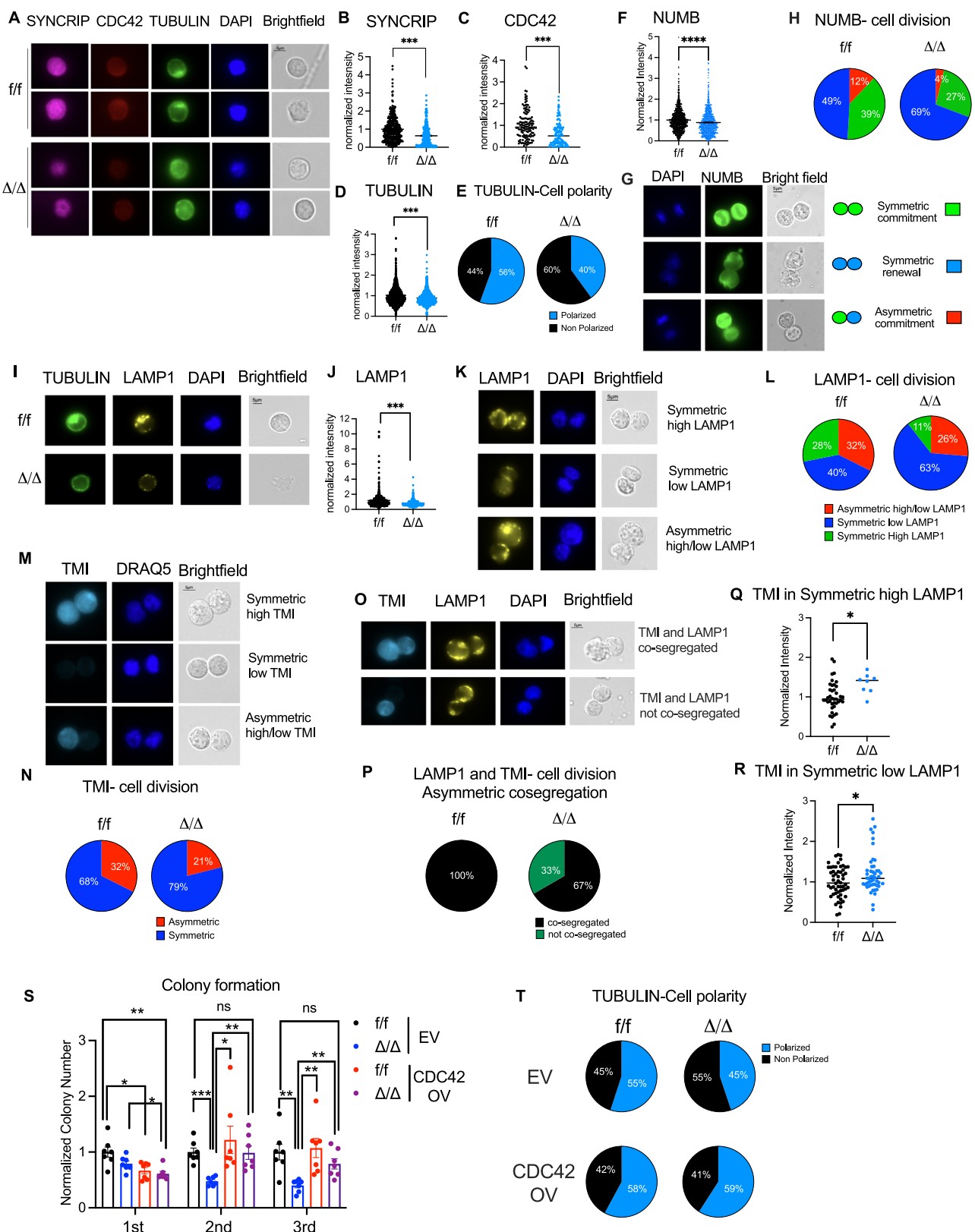

we then wanted to evaluate patterns of NUMB protein in a paired daughter cell division assay. During HSC division, NUMB distribution is used as surrogate indicators for three possibilities: symmetric renewal (both low NUMB), symmetric commitment (both high NUMB) and asymmetric cell division (low vs. high NUMB)[22,52,53]. We found that *Syncrip* deletion reduced frequencies of asymmetric cell division and symmetric commitment with a significant increase in symmetric renewal (Fig. 6G, H).

In addition to NUMB, lysosomal-associated membrane protein 1 (LAMP-1) marked lysosomes[54] can be asymmetrically divided among HSC daughter cells and asymmetric division can dictate cell fate acquisition in these cells[55]. Lysosomes are important degradation machinery in cells, they're responsible for recycling and removal of excessive proteins and other macromolecules[56]. Additionally, it has been shown that HSC division contributes to maintain proteostasis by diluting misfolded proteins, thus allowing HSCs to preserve long-term

**Fig. 6 | SYNCRIP controls HSC polarity and symmetric vs. asymmetric division.**
**A** Representative images of immunofluorescence (IF) staining of SYNCRIP, CDC42 and TUBULIN in hematopoietic stem cells (HSCs). **B**–**D** Quantification of normalized IF intensity SYNCRIP ($Syncrip^{f/f}$ $n = 379$, $Syncrip^{\Delta/\Delta}$ $n = 300$) (**B**) and CDC42 ($Syncrip^{f/f}$ $n = 133$, $Syncrip^{\Delta/\Delta}$ $n = 129$) (**C**) and TUBULIN ($Syncrip^{f/f}$ $n = 2482$, $Syncrip^{\Delta/\Delta}$ $n = 1703$) (**D**). All $p < 0.0001$. **E** Percentage of polarized vs. unpolarized HSCs based on TUBULIN ($n = 4$-5/genotype; $Syncrip^{f/f}$ $n = 99$ and $Syncrip^{\Delta/\Delta}$ $n = 175$).
**F** Quantification of normalized IF NUMB intensity in HSCs. ($Syncrip^{f/f}$ $n = 1319$ and $Syncrip^{\Delta/\Delta}$ $n = 929$, $p < 0.0001$). **G** Representative images of paired NUMB daughter assay of HSCs. **H** Percentage of doublet cells in each type of cell division ($n = 4$-5/genotype; $Syncrip^{f/f}$ $n = 108$ and $Syncrip^{\Delta/\Delta}$ $n = 52$). **I** Representative images of IF staining of TUBLIN, LAMP1 and DAPI in HSCs. **J** Quantification of normalized IF LAMP1 intensity in HSCs. ($Syncrip^{f/f}$ $n = 566$ and $Syncrip^{\Delta/\Delta}$ $n = 566$, $p < 0.0001$).
**K** Representative images of paired LAMP1 daughter assay of HSCs. **L** Percentages of doublet cells in each type of cell division ($n = 4$-5 each genotype; $Syncrip^{f/f}$ $n = 110$

and $Syncrip^{\Delta/\Delta}$ $n = 165$). **M** Representative images of paired TMI daughter assay of HSCs. **N** Percentages of doublet cells in each type of cell division ($n = 4$-5 each genotype; $Syncrip^{f/f}$ $n = 71$ and $Syncrip^{\Delta/\Delta}$ $n = 38$). **O** Representative images of paired TMI and LAMP1 daughter assay of HSCs. **P** Percentages of asymmetric divided cells. ($n = 4$-5 each genotype; $Syncrip^{f/f}$ $n = 10$ and $Syncrip^{\Delta/\Delta}$ $n = 6$). **Q, R** Quantification of normalized IF intensity of TMI in doublet cells symmetric high LAMP1 $Syncrip^{f/f}$ $n = 40$ and $Syncrip^{\Delta/\Delta}$ $n = 8$, $p = 0.018$ (**Q**) and symmetric low LAMP1 $Syncrip^{f/f}$ $n = 56$ and $Syncrip^{\Delta/\Delta}$ $n = 48$, $p = 0.022$ (**R**). **S** Normalized colony numbers 1st, 2nd and 3rd plating of $Syncrip^{f/f}$ and $Syncrip^{\Delta/\Delta}$ LSK cells transduced with empty vector (EV) or expressing CDC42 (CDC42-OV) ($n = 7$/condition). **T** Percentages of polarized vs. unpolarized cells based on TUBULIN ($Syncrip^{f/f}$-EV $n = 20$; $Syncrip^{\Delta/\Delta}$-EV $n = 29$; $Syncrip^{f/f}$ CDC42-OV $n = 22$; and $Syncrip^{\Delta/\Delta}$ CDC42-OV $n = 19$). Scale bars 5µm. Source data are provided as Source Data File. All data represent mean ± s.e.m. p values were calculated by two-tailed $t$-test. $^*p < 0.05$, $^{**}p < 0.01$, $^{***}p < 0.001$ and ns: not significant.

regenerative capacity[15]. Hence, lysosome activity and distribution could play an important role in mediating these processes. Moreover, generation and localization of lysosomes and associated vesicles i.e., autophagosomes and mitophagosomes is highly dependent on the transport of vesicular cargos and intracellular trafficking mediated by dynamic activities and polarization of the cytoskeleton[57–59]. Thus, we examined lysosomes in HSCs upon SYNCRIP depletion and found that LAMP1 abundance was reduced compared to the controls (Fig. 6I, J). Interestingly, we observed a shift to increased symmetric division in cells with low LAMP1 expression at the expense of reduced asymmetric division of LAMP1 partition in $Syncrip$ deficient HSCs (Fig. 6K, L). Altogether, these data suggest that SYNCRIP regulates cell polarization, and partition of lysosomes possibly via its control of GTPase-mediated cytoskeleton activities in HSCs.

Given our observation of increased unfolded proteins in $Syncrip$ deficient HSCs (Fig. 4), we wanted to explore a potential connection between altered lysosomes and segregation of unfolded proteins during HSC division. We first confirmed that TMI is elevated in SYNCRIP deleted HSCs by IF staining (Fig. S6A). Interestingly, we observed that TMI levels in HSCs were strongly correlated with LAMP1 and NUMB protein levels (Figure S6B, D), suggesting that lysosomal production might correspond to accumulation of TMI marked unfolded proteins. Therefore, we asked whether TMI distribution to HSC daughter cells during division follows the pattern of LAMP1 and NUMB commitment divisions. We observed that TMI was also divided into paired daughter cells in three distinct patterns i.e. symmetric high TMI, symmetric low TMI and asymmetric high/low TMI (Fig. 6M) analogous to LAMP1 and NUMB. $Syncrip$ deletion resulted in fewer HSCs undergoing TMI asymmetric division (Fig. 6N).

Furthermore, co-staining of HSCs with TMI and LAMP1 (Fig. 6O and S6E) demonstrated that LAMP1 and TMI co-segregated when LAMP1 and TMI were asymmetrically divided in paired daughter cells. Upon SYNCRIP depletion, the pattern of co-segregation of LAMP1 and TMI was disrupted (Fig. 6P). Interestingly, while there was no significant shift in other co-staining patterns of LAMP1 and TMI, we found a significant an increase in TMI levels in HSCs undergoing both symmetric high or low LAMP1 but not asymmetric division (Fig. 6Q–R and S6F). Elevation in TMI signals were also observed when co-staining TMI with NUMB in the daughter cell assay (Fig. S6G–J). These data suggested that loss of $Syncrip$ resulted in defective lysosomal-mediated degradation and partition of misfolded proteins. This in turn may affect the ability of HSCs to dump out misfolded proteins to progenitor cells during asymmetric division, thus further exacerbating the stress within HSCs.

To further demonstrate CDC42 as the functional downstream target of SYNCRIP in HSCs, we evaluated whether restoration of CDC42 can rescue the defects in $Syncrip$ deficient cells. We retrovirally overexpressed CDC42 in sorted LSK cells from $Syncrip$ KO and $Syncrip$ WT mice and observed that CDC42 overexpression at least partially

reversed the reduction in ability of HSPCs to serially replate after SYNCRIP depletion (Fig. 6S). Importantly, overexpression of CDC42 also restored TUBULIN abundance (Fig. S6K, L) and reestablished TUBULIN-dependent cell polarity in SYNCRIP deleted HSPCs (Fig. 6T, Supplemental Movie 1, 2, 3, 4).

To test if CDC42 overexpression can rescue the defect in vivo, $Syncrip^{\Delta/\Delta}$ and $Syncrip^{f/f}$ LSK cells from primary mice were transduced with CDC42 overexpression (OV) retrovirus, and $1 \times 10^4$ transduced LSK cells were transplanted into CD45.1 recipients. Chimerism in the bone marrow was evaluated at 24 weeks post-transplantation to read out long-term engraftment. We observed that CDC42 OV did not improve SYNCRIP KO HSPCs' engraftment deficiency at the level of total bone marrow and multiple progenitors. However, OV of CDC42 resulted in preservation of long-term engraftment within the LT-HSCs (Fig. S6M). The data suggested that while CDC42 alone is not sufficient to rescue the full reconstitution potential of SYNCRIP-deficient HSPCs, CDC42 at least in part is responsible for SYNCRIP's role in maintaining reconstitution in HSCs. The data also suggest that there might be context-dependent targets and function for SYNCRIP between HSCs vs. HSPCs and possibly downstream hematopoietic cells. Our multi-omics analysis in fact identified many other targets and pathways influenced by SYNCRIP, including other Rho GTPases and other RBPs. It is conceivable that in the context of progenitors and total bone marrow engraftment, activities of multiple targets are required to compensate for loss of $Syncrip$. Overall, these data suggest that SYNCRIP regulated HSCs' function in part through translational control of CDC42.

## Discussion

RNA binding proteins are emerging as key regulators in maintaining HSCs by impacting properties such as self-renewal and cell fate decisions[22,34,60–62]. Here, our study demonstrates that the RBP SYNCRIP, which has previously been identified to be required for the leukemia stem cell program[24] maintains self-renewal of HSPCs in adult hematopoiesis. Upon conditional ablation of $Syncrip$, we found reduced lymphocyte blood counts in the periphery and an increase in LSK frequency. The data indicates an overall modest effect on adult blood cell development and steady state hematopoiesis. The most significant defects observed were reduced reconstitution capacity of the HSCs and progenitors emerged from various regenerative insults including transplantation and irradiation.

It was previously identified that protein homeostasis is maintained in a cell-specific manner, where HSCs have lower protein synthesis rates than restricted progenitors[13]. HSCs are particularly sensitive to alterations in protein synthesis and even a modest increase in protein synthesis reduces HSCs protein quality, resulting in an impairment of self-renewal[15,63]. Transcriptional profiling of our SYNCRIP KO HSCs demonstrated an increase in stem cell and translation signatures, along with an increase in nascent global protein synthesis.

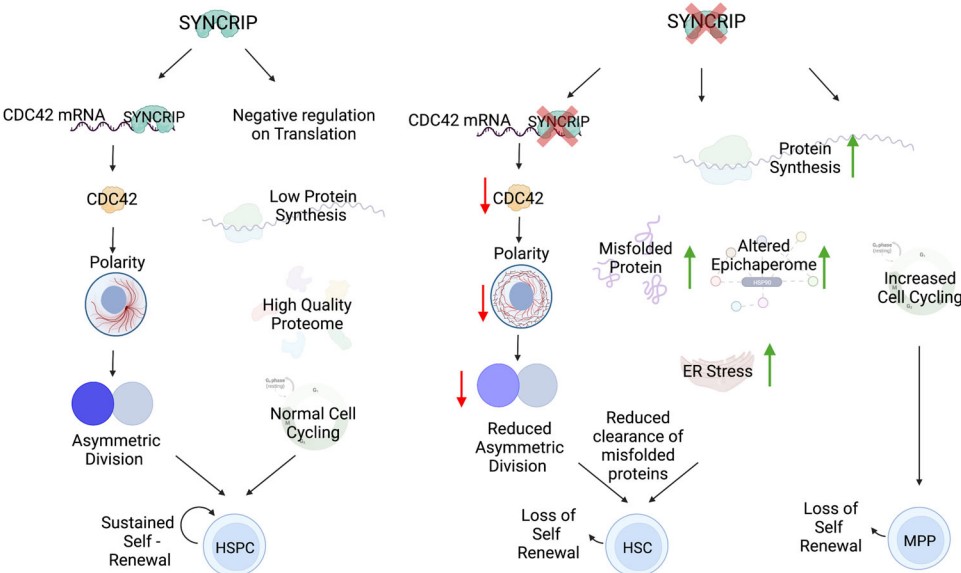

**Fig. 7 | Graphical abstract illustrating SYNCRIP controls proteome quality and CDC42-mediated cell polarity and division to maintain self-renewal of reserve HSCs.** Model showing mechanism for how SYNCRIP regulates HSPC properties. Left panel shows normal conditions with SYNCRIP and right panel shows HSPC conditions without SYNCRIP. Figure created with BioRender.com.

We also observed that SYNCRIP KO HSCs have an increased burden of unfolded/misfolded proteins. These downstream effects after *Syncrip* loss likely contribute towards a failure to maintain HSC self-renewal. On the other hand, it is noted that we also observed an increase in cycling MPPs after transplantation, suggesting that activation of progenitor compartments and subsequently reduction of blood production may contribute to the overall impact on HSPC function. Although loss of quiescence can lead to loss of self-renewal, we found that transplanted *Syncrip* deficient HSCs exhibited equivalent cycling, activation, and division with WT HSCs.

There are several pathways that HSCs utilize to protect themselves from proteotoxic stress such as the ubiquitin-proteasome system[64], autophagy[65,66], and the ER unfolded protein response[67,68]. The purpose of these pathways is to relieve stressors by attenuating translation, removing or promoting proper folding of unfolded proteins, and, in the case of chronic stress, inducing apoptosis in HSCs[69]. Single-cell transcriptomic (scRNA-seq) analysis demonstrated a deregulated stress response unique to SYNCRIP KO HSC populations, which was characterized by HSF-1-dependent activation of heat shock proteins i.e., HSP70 and HSP90. We further validated increased expression of HSP70 protein as well as an accumulation of a dysregulated epichaperome network that was more sensitive to inhibition of HSP90 in HSCs. Despite being previously characterized in cancer, including acute myeloid leukemia as a pro-survival mechanism and a response to oncogenic stress[39,70], the formation of the aberrantly reorganized chaperone protein networks reflects an abnormal cellular state. Therefore, the enhanced epichaperome observed in *Syncrip* deficient HSCs strongly indicated an altered proteome.

Additionally, while HSF1 activation has been shown to help maintain proteostasis in cultured and aging HSCs[71], activation of HSF1 response in SYNCRIP KO HSCs is not sufficient to compensate for SYNCRIP's loss of function. In fact, we also observed activation of ER stress, specifically an increase in ATF4 and CHOP protein expression associated with the PERK arm of the UPR, as well as morphological changes in the ER that suggest the presence of mild ER stress. As previously demonstrated, the altered stress program was selective for HSCs and not MPPs. Despite activation of these stress response pathways, SYNCRIP KO HSCs failed to alleviate the proteotoxic stress that likely contributes to HSC function. This suggests a broad and

essential role of SYNCRIP in safeguarding the healthy proteome in HSCs.

Although we did not observe stress-related pathways in the MPPs, we did observe upregulation of multiple mitochondrial and respiratory electron transport-related pathways indicating SYNCRIP may be influencing mitochondrial activity in the MPP populations. Multiple studies have linked the importance of mitochondrial activity with HSC function, and it's been shown that lower mitochondrial activity promotes self-renewal function in HSCs[72,73]. Further work will be necessary to address what role SYNCRIP plays in influencing mitochondrial activity.

To identify the direct effectors and downstream functional pathways of SYNCRIP in HSCs, we employed a multiomic approach where we incorporated techniques tailored for low input materials derived from HSCs. We utilized Hypertribe technology to uncover SYNCRIP's direct mRNA targets and transcriptomic and proteomic profiling of HSCs and HSPCs respectively. We found that direct binding of SYNCRIP to its mRNA targets generally does not impact its abundancy, strongly suggesting that SYNCRIP might play a more dominant role in post-transcriptional gene expression regulation. We identified that SYNCRIP has two major direct effects (Fig. 7). The first is maintaining translation of other RNA binding proteins including MSI2 and other RBPs that control the low translational output in HSCs. Secondly, we uncovered a direct link between SYNCRIP with several functional pathways associated with GTPase activity and cytoskeleton organization. We found that SYNCRIP translationally controls expression of Rho GTPase CDC42, which is well characterized to be essential for HSC engraftment[48,49,74] and it has been observed that changes in CDC42 activity or expression creates cytoskeletal polarity defects in HSCs[49,75,76].

The establishment of cell polarity by CDC42 also helps regulate cell division and segregation of cell fate determinants and dilution of protein components[77]. It has been recently described that lysosome are asymmetrically inherited with asymmetrical inheritance of lysosomes predicting differentiated multipotent progenitor cells[55]. Lower lysosomal inheritance is associated with co-inheritance of differentiation markers such as CD71, suggesting cells with HSC daughter cells with low levels of lysosomes are primed to differentiate[55]. Additionally, HSC quiescence has been linked to greater abundance of large

lysosomes[78]. We observed SYNCRIP depletion decreasing cell polarization, LAMP-1 marked lysosomes and asymmetric inheritance of lysosomes. Another role for polarization and asymmetric cell division is to deposit misfolded proteins into the differentiated daughter cells protecting the HSCs (Hidalgo San Jose et al., 2020). In fact, we observed both elevated accumulation of TMI in HSCs and abnormal pattern of distribution of TMI together with LAMP1 and NUMB in paired daughter cells. This suggest that there is a failure to remove excessive misfolded proteins in HSCs due to inability to deposit them to differentiated daughter cells. These abnormal cellular activities we proposed impact the ability to maintain the self-renewal capacity of HSCs. Due to the challenges to assess in vivo asymmetric division it is difficult to fully prove if these defects in polarity are driving the in vivo self-renewal loss.

In summary, we demonstrate that SYNCRIP has an essential role in maintaining self-renewal of HSPCs. We also demonstrate a mechanism for how an RBP maintains HSC proteostasis, advancing the current understanding of how RBPs and post-transcriptional regulation influence stem cell properties.

## Methods

### Animal research ethical regulation statement
All animal studies were performed on animal protocol #11-10-025 approved by the Institutional Animal Care and Use Committee (IACUC) at Memorial Sloan Kettering Cancer Center.

### Mouse transplants
The MSKCC animal facility is maintained at temperatures between 64–78° F, with humidity of the animal room ranging between 30-70%. In the noncompetitive primary transplant $2 \times 10^6$ whole bone marrow cells from 6–8-week-old *Syncrip f/f*, Mx1-Cre+ or Mx1-Cre- mice (strain developed at MSKCC) were transplanted into lethally irradiated 6–8-week-old female B6SJL congenic CD45.1 recipients (Taconic Biosciences, stock #4007). In the competitive primary transplants $10^6$ whole bone marrow cells from 6–8-week-old *Syncrip* cKO mice and $10^6$ cells from 6–8-week-old female congenic CD45.1 (Taconic Biosciences, stock #4007) mice were injected into 6-8 week old female CD45.1 (Taconic Biosciences, stock #4007) recipients. In the cell-autonomous transplants we transplant $2 \times 10^6$ whole bone marrow cells from 6–8-week-old *Syncrip f/f*, Mx1-Cre+ or Mx1-Cre- into congenic 6–8-week-old female CD45.1 recipient mice (Taconic Biosciences, stock #4007). Mice are bled at 5 weeks post-transplant to ensure engraftment, and after 6 weeks of engraftment mice deletion is induced via pIpC (InVivogen, Vac-Pic) intraperitoneal injections at a dose of 10 mg/kg on 2 consecutive days[34]. Sex was considered in the study design such that phenotypic assessments in primary animals were performed using both male and female mice. Donor cells for primary transplantation were taken from both male and female animals. There was no observable difference between sexes.

### Western blot
*Syncrip*[f/f] and *Syncrip*[Δ/Δ] bone marrow cells were harvested; 250,000 cells were then lysed in 40 uL 1x Laemmli protein loading buffer and boiled for 5 min. Whole cell lysates were run on 4%–15% gradient SDS-PAGE and transferred to nitrocellulose membrane. Membranes were probed with the SYNCRP (Millipore Sigma, MAB11004, dilution 1:1000) and ACTIN (Sigma Aldrich, A3854, dilution 1:20000) antibody.

### Flow cytometry and cell sorting
Flow cytometry and FACS were performed as follows and based on protocols in[22,34]. Bone marrow cells were isolated and subjected to red blood cell (RBC) lysis. To measure the HSPC compartments, cells were stained with the following cocktail of flow cytometry antibodies: Lineage markers (CD3 (Invitrogen, Cat#15-0031-83), CD4 (Invitrogen, Cat#15-0041-83), CD8 (Invitrogen, Cat#15-0081-83), Gr1 (Invitrogen,

Cat#15-5931-82), B220 (Invitrogen, Cat#15-0452-83), CD19 (Thermo Fisher Scientific, Cat#15-0193-83) and Ter119 (Thermo Fisher Scientific, Cat#15-5921-82)) – PE Cy5, cKit-APC Cy7 (Biolegend, Cat#105826), Sca-1-Pacific Blue (Biolegend, Cat#122520), CD150-APC (Biolegend, Cat#115910), CD48-PE (BD Biosciences, Cat#557485). To monitor linage cells differentiation, cells were stained with cocktail including Gr1-APC (Thermo Fisher Scientific, Cat#17-5931-82), Mac1-Pacific Blue (Biolegend, Cat#101224), Ter119-PE Cy5 (Thermo Fisher Scientific, Cat#15-5921-82), CD71-FITC (Thermo Fisher Scientific, Cat#11-0711-81), CD41-PE (BD Biosciences, Cat#557308), or a cocktail containing CD3-Pacific Blue (Thermo Fisher Scientific, Cat# 48-0031), CD4-PE (BD Biosciences, Cat# 558040), B220-PE Cy7(BD Biosciences, Cat#552772), CD19-PE Cy5 (Thermo Fisher Scientific, Cat#15-0193-83), IgM-APC (Biolegend, Cat#406509), CD43-FITC(BD Biosciences, Cat#553270). Stem/progenitor populations and mature lineages are defined as HSC – hematopoietic stem cells (Lin-Sca+ckit + (LSK) CD48-CD150 + ); MPP – multipotential progenitor (LSK-CD48 + /-CD150-); MPP1-LSK CD48-CD150-; MPP2-LSK CD48 + CD150 + ; MPP4-LSK CD48 + CD150-. CMP – common myeloid progenitor (Lin-Sca+ckit- CD34 + FcγRII/III mid); GMP – granulocyte monocyte progenitor (Lin-Sca+ckit- CD34 + FcγRII/III high); MEP – megakaryocyte erythrocyte progenitor (Lin-Sca+ckit- CD34- FcγRII/III low); myeloid – (Mac1 + Gr1 + ); B cells (B220 + ) and T cells (CD3 + ).

For transplanted mice, we added CD45.1-PE Texas Red (BD Biosciences, Cat#562452) and CD45.2-A700 (Thermo Fisher Scientific, Cat#56-0454-82) to distinguish donor and recipient cells. For HSPC (LSK) or HSC (LSK, CD150 + CD48-) cell sorting, bone marrow cells were harvested and incubated with 50uL CD117 MACS beads (Miltenyi Biotec, Cat#130-091-224) for 30 min, then run on AutoMACS according to manufacturer's instructions. Cells were then stained with antibody cocktail: Lineage marker (CD3 (Invitrogen, Cat#15-0031-83), CD4 (Invitrogen, Cat#15-0041-83), CD8 (Invitrogen, Cat#15-0081-83), Gr1 (Invitrogen, Cat#15-5931-82), B220 (Invitrogen, Cat#15-0452-83), CD19 (Thermo Fisher Scientific, Cat#15-0193-83), Ter119 (Thermo Fisher Scientific, Cat#15-5921-82))-PE Cy5, cKit-APC Cy7 (Biolegend, Cat#105826), Sca-1-Pacific Blue (Biolegend, Cat#122520), CD150-APC (Biolegend, Cat#115910), CD48-PE (BD Biosciences, Cat#557485), and included CD45.1-PE Texas Red (BD Biosciences, Cat#562452), CD45.2-A700 (Thermo Fisher Scientific, Cat#56-0454-82) for transplanted mice. Specific cell populations were sorted on BD Aria instrument. All flow cytometry and FACS antibody were used at 1:200 dilutions.

### Colony formation assay
10000 whole *Syncrip*[f/f] or *Syncrip*[Δ/Δ] bone marrow cells were plated in M3434 methylcellulose media, colonies were scored at 7 days post-plating. For PU-H71 treatments and CDC42 Rescue, 750 LSK (Lin-Sca1+cKit + ) cells were plated in methylcellulose media for the first plating, and colonies were scored at 7 days post-plating. Subsequent replatings were done with 5000 cells and colonies were scored 7 days post-plating.

### Immunofluorescence
HSCs were sorted from *Syncrip*[f/f] and *Syncrip*[Δ/Δ]mice following protocol shown above. We performed daughter cell assays modifying protocols used in[22,34], in order to do daughter paired assays for multiple markers. For NUMB daughter cell assay, sorted HSCs were cultured with SFEM media containing 10 ng/mL SCF and 20 ng/mL TPO in 96 round-bottom wells for 16–18 h and then treated cells with 10 nM Nocodazole for 24 h. After incubation cells were fixed with 1.6% paraformaldehyde 15 min at room temperature and permeabilized with ice-cold methanol. Fixed HSCs were then cytospun onto poly-L-lysine coated glass slides and blocked for 1 h with PBS + 0.5%BSA. Slides were then stained with anti-Numb (Abcam, Cat#ab4147, dilution 1:500), anti-SYNCRIP (Millipore Sigma, Cat#MAB11004, dilution 1:500), and secondary Ab (Alexa Fluor 488 donkey anti-goat, Invitrogen Cat#A11055, dilution

1:500; Alexa Fluor 647 donkey anti-mouse, Invitrogen Cat#A31571, dilution 1:500), followed by DAPI counterstaining (Hoechst, Thermo Fisher Scientific Cat#H3570, dilution 1:1500). We evaluate symmetric and asymmetric percentages based on the fluorescence signal intensity of each cell acquired by Axio Imager M2 microscope (Carl Zeiss) with a 63X objective lens, using the Zeiss Zen microscopy software for data acquisition and quantified images by ImageJ software (Version 2.0.0). Thresholds to determine NUMB high/low expression were set for individual experimental replicates. To score NUMB pairs, intensity ratios were assessed, if there was at least a 2-fold difference (or greater) between daughter cells, the pair was scored as asymmetric. All pairs with a less than 2-fold difference were scored as symmetric, with high average intensity NUMB expression being scored as symmetric commitment. Symmetric pairs with low average intensity NUMB expression were scored as symmetric renewal.[22,34] In a separate experiment, HSC cells were stained for SYNCRIP (Millipore Sigma, Cat#MAB11004, dilution 1:500), CDC42 (Abcam, Cat#ab64533, dilution 1:500), and TUBULIN (Abcam, Cat#ab6160, dilution 1:500) using secondary Ab donkey anti-mouse Alexa Fluor 647 (Invitrogen Cat#A31571, dilution 1:500), donkey anti-rabbit Alexa Fluor 568 (Invitrogen, Cat#A10042, dilution 1:500), and goat anti-rat Alexa Fluor 488 respectively (Invitrogen, Cat#A11006, dilution 1:500). Separately, we also stained HSC for TUBULIN (Abcam, Cat#ab6160, dilution 1:500) and LAMP1 (Abcam, Cat#ab208943, dilution 1:500) using secondary Ab goat anti-rat Alexa Fluor 488 (Invitrogen, Cat#A11006, dilution 1:500) and donkey anti-rabbit Alexa Fluor 568 respectively (Invitrogen, Cat#A10042, dilution 1:500).

For LAMP1 daughter cell assay, cell pairs were scored as asymmetric if their intensities had a Log2FC ≥ ±0.6, while pairs with Log2FC < ±0.6 intensities were scored as symmetric. A separate batch of HSCs was stained for ATF4 (Cell Signaling Technology, Cat#11815, dilution 1:100) and CHOP (Cell Signaling Technology, Cat#2895, dilution, 1:100) using secondary Ab donkey anti-rabbit Alexa Fluor 568 (Invitrogen, Cat#A10042, dilution 1:500) and donkey anti-mouse Alexa Fluor 647 (Invitrogen, Cat#A31571, dilution 1:500) respectively.

For unfolded protein IF daughter assay, HSCs were sorted and treated with nocodazole as described above. Cells were then live stained with Tetraphenylethene maleimide (TMI) at a final concentration of 50 uM in PBS for 45 min at 37 °C and subsequently fixed with 1.6% paraformaldehyde and permeabilized with ice-cold methanol same as described above. Cells were then co-stained with NUMB (Abcam, Cat#ab4147, dilution 1:500) and LAMP1(Abcam, Cat#ab208943, dilution 1:500) followed by secondary Ab (Alexa Fluor 488 donkey anti-goat, Invitrogen Cat#A11055, dilution 1:500; Alexa Fluor 568 donkey anti-rabbit, Invitrogen Cat#A10042 dilution 1:500), and lastly with DRAQ5 for nuclear staining (Abcam, Cat# ab108410, dilution 1:1000). Cells were imaged on Axio Imager M2 microscope (Carl Zeiss) same as described above and NUMB and LAMP1 were scored same as described above. TMI asymmetry was scored in a similar manner as LAMP1, if pair intensities had Log2FC ≥ ±0.6 cells were scored as asymmetric and if pair intensities had Log2FC < ±0.6 then they were scored as symmetric.

For CDC42 Rescue, LSK cells were sorted and transduced. 48 h post-transduction cells were fixed in 1.6% paraformaldehyde for 15 min at room temperature then permeabilized with ice-cold methanol. Cells expressed GFP and were stained with TUBULIN (Abcam, Cat#ab6160, dilution 1:500) and CDC42 (Abcam, Cat#ab64533, dilution 1:500) followed by secondary Ab (Alexa Fluor 568 goat anti-rat, Invitrogen Cat#A11077, dilution 1:500, Alexa Fluor 647 donkey anti-rabbit, Invitrogen Cat# A-31573) and DAPI (Hoechst, Thermo Fisher Scientific Cat#H3570, dilution 1:1500). Cells were imaged on confocal microscope (Leica TCS SP5 II in an upright configuration) utilizing a x63 objective and images were quantified by FIJI. To score polarity, integrated intensity was determined across all z-stacks and cells were split

in half. If the TUBULIN intensity was Log2FC > 0.6 between the two halves, then that cell was scored as polar.

## Retroviral production

Virus was produced in HEK 293 T cells using pCL-Eco (Addgene #12371) and pMD2.G (Addgene #12259). SYNCRIP-ADAR and CDC42 were separately cloned into MSCV-IRES-GFP (MIGR1, Addgene #27490) and used to generate retrovirus. HEK 293 T cells were transfected with 0.25 M CaCl₂, and BES buffered saline solution. Media was changed 12–16 h post transfection and the first virus suspension was collected 24hrs post media change. A second collection was harvested 48 h post media change. Concentrated stocks of virus were prepared by precipitating virus using 50% PEG 6000 (poly(ethylene glycol)) and 4 M NaCl for 2 h, with inversion every 20 min. Precipitates were pelleted down at 2000 x g for 30 min at 4 °C. Virus pellets were vigorously resuspended in HBSS (without Ca, Mg, and phenol red), suspension was pelleted down again. Supernatant was collected and used for retroviral transductions.

## HSPC retroviral transduction

HSPCs were sorted as described above and cultured with SFEM media supplemented with murine cytokines (50 ng/ml SCF, 10 ng/ml IL-3, and 10 ng/ml IL-6, 10 ng/ml TPO and 20 ng/ml FLT3L). Cells were transduced with retroviral suspensions in the presence of 10ug/mL polybrene and followed with spin infection for 1 h. After 24 h, cells were transduced again as previously described. 48 h later, cells were FACS sorted for GFP expression using BD Aria instrument.

## O-Propargyl-puromycin (OP-Puro) flow analysis

Protein synthesis was assessed by O-Propargyl-puromycin (OP-Puro) using the Click-iT® Plus OPP Alexa Fluor 594 Protein Synthesis Assay Kit, following the manufacturer's instructions. Control cells were treated with 150 ug/mL cycloheximide for 15 min. *Syncrip* f/f and *Syncrip* Δ/Δ HSC cells were sorted and then treated with 30 uM Click-iT OPP Reagent (component A) for 1 h. Cells were then washed and fixed in 1.6% paraformaldehyde for 15 min room temperature, and permeabilized in ice-cold methanol. Labeled cells were analyzed using a BD Fortessa instrument.

## Measurement of unfolded proteins

To measure unfolded protein, *Syncrip* f/f and *Syncrip* Δ/Δ bone marrow cells were isolated, lysed with RBC lysis buffer twice and washed twice with Ca2 + - and Mg2 + -free PBS. 6 × 10⁶ *Syncrip* f/f and *Syncrip* Δ/Δ cells were treated with Tetraphenylethene maleimide (TMI; stock 2 mM in DMSO). TMI was diluted in PBS (50 uM final concentration) and added to each sample, The samples were then incubated at 37 °C for 45 min. Samples were washed twice in PBS and then stained with flow cytometry cell surface markers as described above. Cells were analyzed using a BD Fortessa instrument.

## Transmission electron microscopy (TEM)

Mouse bone marrow was isolated and ~9 × 10⁴ HSCs were sorted as described above. Samples were washed with PBS then fixed with a modified Karmovsky's fix of 2.5% glutaraldehyde, 4% parafomaldehye and 0.02% picric acid in 0.1 M sodium caocdylate buffer at pH 7.2[79]. Following a secondary fixation in 1% osmium tetroxide, 1.5% potassium ferricyanide[80] J Ultrastruct. Res 42:29, samples were dehydrated through a graded ethanol series, and embedded in an epon analog resin. Ultrathin sections were cut using a Diatome diamond knife (Diatome, USA, Hatfield, PA) on a Leica Ultractu S ultramicrotome (Leica, Vienna, Austria). Sections were collected on copper grids and further contrasted with lead citrate[81] and viewed on a JEM 1400 electron microscope (JEOL, USA, Inc., Peabody, MA) operated at 100 kV. Images were recorded with a Veleta 2 K x2K digital camera (Olympus-SIS, Germany). Images were quantified on Fiji.

## Epichaperome detection flow cytometry assay

The epichaperome probes, PU FITC and FITC9, were synthesized as described in ref. 82. And the assay was performed as previously described[39],[82]. Mouse whole bone marrow was isolated, lysed twice with RBC lysis buffer and washed with FACS buffer (PBS + 2% FBS + 5 mM EDTA). $6 \times 10^6$ bone marrow cells per condition were incubated with 1 uM PU FITC at 37 °C for 4 h. Cells were then washed three times with FACS buffer and stained with flow cytometry cell surface markers as described above. Cells were analyzed using a BD Fortessa instrument. The FITC derivative FITC9 was used as a negative control, and the fold PU FITC binding was calculated relative to FITC9 control.

## Single-cell RNA-seq analysis

Single-cell RNA-seq data was processed and aligned to mouse reference mm10 using CellRanger with default parameters. Normalization, dimensionality reduction and Louvain clustering were performed using Scanpy (v1.4.4[83]. Cells with <200 detected genes or percentage of mitochondrial reads >15% were filtered. Data were log normalized using a scale factor of 10000. Louvain clusters were assigned the cell type with the best match by scoring expression of marker genes of clusters from[34]. Diffusion maps of early hematopoiesis were calculated by limiting to cells in clusters assigned to cell types: HSC-C1, HSC-C2, MPP1, MPP2[84]. Partition-based abstracted graph (PAGA) representation was calculated separately on each condition to highlight differences in connectivity[35].

## In-gel digestion and mass spectrometry

All reagent used where mass spectrometry grade. Proteomes were resolved by SDS-PAGE 10% polyacrylamide Bis-Tris gels (Invitrogen) at 100 V (~30 min).To visualize proteins, gel was stained using Silver Stain for Mass Spectrometry kit (Pierce) according to manufacturer's instructions. Relevant gel portions were excised and destained using 50 µl of 30 mM $K_3[Fe(CN)_6]$ in 100 mM $Na_2S_2O_3$ by incubation at room temperature for 30 mins, under constant agitation at 700 rpm. Following destaining, 500 µl 25 mM $NH_4HCO_3$ (ABC) was added to each tube and incubated for 5 min at room temperature under constant agitation. Solution was removed and gel pieces were washed trice for 10 min with 500 µl 50% acetonitrile in 25 mM aqueous ABC. All solution was removed and 100 µl acetonitrile was added to each tube and incubated for 5 min at room temperature under constant agitation, followed by lyophilization using a vacuum centrifuge. Gel slabs were re-hydrated with 25 µl of 10 mM dithiothreitol (DTT) in 100 mM ABC (1 h at 56 °C), followed by alkylation with 25 µl of 55 mM iodoacetamide in 100 mM ABC (30 min, room temperature in the dark), and quenching with 5 microliters of 100 mM DTT in 100 mM ABC (5 min at room temperature). Gel fragments were washed trice by adding 50 µl of acetonitrile and incubating for 5 min at room temperature, followed by addition of 500 µl 100 mM ABC and incubation for 10 min at room temperature. After a final wash with 100 µl acetonitrile for 10 min at room temperature, gel fragments were lyophilized. Gel slabs were reconstituted in 50 µl of 12.5 ng/µl Sequencing Grade Modified trypsin (Promega) in 50 mM ABC, and proteolysis was performed for 16 h at 37 °C. Peptides were eluted twice by incubating the gel slabs for 30 min with 50 µl 1% aqueous formic acid in 70% acetonitrile under constant agitation at 1400 rpm. Eluates were pooled, lyophilized, and stored at −80 °C until analysis. Samples were resuspended in 20 µl 0.1% aqueous formic acid, and 3 µl were analyzed by LC-MS. The LC system consisted of a vented trap-elute configuration (Ekspert NanoLC 425 chromatograph, Eksigent) coupled to a Orbitrap Fusion mass spectrometer (Thermo Fisher Scientific) via a nano electro-spray DPV-565 PicoView ion source (New Objective), as previously described[85]. After being loaded on the trap column (5 cm × 150 µm ID, packed with Poros R2-C18 10 µm particles (Life Technologies)) peptides were resolved on a nanoscale reversed phase analytical column (60 cm × 75 µm ID,

packed with ReproSil-Pur C18-AQ 30 µm particles (Dr. Maisch), and kept at constant 55 °C), using a 180 min 3–40% linear gradient of acetonitrile/ 0.1% formic acid (buffer B) in water/0.1% formic acid (buffer A) at 300 nL/minute. Eluted peptides were transferred in gas phase by electrospray ionization using a 3 µm ID silica emitter with potential decreasing from 1800 V to 1600 V in 50 V steps over the elution. Precursor ions in the 385–1600 m/z range were isolated using the quadrupole and recorded every 3 s using the Orbitrap detector (120,000 resolution), with an automatic gain control target set at $10^6$ ions and a maximum injection time of 50 ms. The Targeted Mass trigger was applied to bias precursor selection towards the ions predicted to be generated from targeted proteins (Supplementary Data 15), with a mass tolerance of 10 ppm. For each targeted protein, tryptic peptides and charge states were predicted using the Skyline[86] software for up to 3 unique peptides per proteins, based on consensus sequences from mouse UniProt database as of October 2019[87]. If no targeted trigger ions were detected, the mass spectrometer was set to perform data-dependent MS2 precursor selection, limiting fragmentation to monoisotopic ions with charge 2–4, and dynamically excluding already fragmented precursors for 30 s (10 ppm tolerance). Selected precursors were isolated (Q1 isolation window 1.2 Th) and HCD fragmentated (stepped normalized collision energy 26-32-38%) using the top speed algorithm. Product ion spectra were recorded in the orbitrap at 30,000 resolutions (AGC $5 \times 10^4$ ions, maximum injection time 32 ms), in centroid mode.

The mass spectrometry proteomics data have been deposited to the ProteomeXchange Consortium via the PRIDE[88] partner repository with the dataset identifier PXD019460. Mass spectra were analyzed using Peaks Studio version 10.5 (BSI) For identification, spectra were matched against the murine UniProt database (as of October 2019), supplemented with contaminant proteins from the cRAP database[89] with FDR < 0.01. Mass tolerance was set at 10 ppm and 0.1 Da for precursor and fragment ions, respectively. Cysteine carbamidomethylation was set as fixed chemical modification, while methionine oxidation, NQ deamidation and protein N-terminus acetylation were set as variable. Protease specificity was set to trypsin, with up to 3 missed cleavages allowed (max 3 PTM per peptide. The match between runs feature was enabled (0.7 min tolerance, 20 min alignment). Quantification was performed using the LFQ algorithm using the area under the curve defined by the extracted ion chromatogram for each peptide precursor ion as quantitative metric. Raw protein intensities (i.e. the sum of the max intensity values for all peptides matched to a given protein) were extracted from the proteinGroups.txt table and used as quantitative metric. After removing obvious contaminant proteins (mapping in the cRAP database) and proteins with no quantification, intensities were normalized to equalize the total intensity for all 6 samples. Normalized protein intensities were used to calculate log2 transformed ratios and significance of differential quantification. A one-sided Wilcoxon test was performed to determine the statistical significance between the log2 fold changes (log2FC).

## CFSE staining for homing and in vivo proliferation

To test homing capacities, LSK cells were sorted as previously described. Cells were stained with 3uM CFSE in PBS for 7 min at 37 °C in the dark. $1.8 \times 10^4$ CFSE stained LSK cells were transplanted into lethally irradiated congenic CD45.1 recipients. Mice were sacrificed 16hrs after transplantation and the BM from femora, tibiae and pelvis was collected and evaluated using flow cytometry as previously described. For in vivo proliferation, LSK cells were sorted as described above and stained with 3uM CFSE in PBS for 7 min at 37 °C in the dark. $3.2 \times 10^4$ CFSE stained cells were transplanted into lethally irradiated congenic CD45.1 recipients. Mice were sacrificed 1 week post-transplant. BM from femora, tibiae and pelvis were collected and evaluated using flow cytometry as previously described.

### In vitro live cell imaging

HSC cells were sorted as previously described. All live imaging was conducted on the CellRaft AIR System (Cell Microsystems) according to the manufacturer's instructions. Briefly, the raft array was prepared by rinsing the reservoir with warm PBS and incubating for 3 min at room temperature. This process was repeated 3 times, without letting the wells dry. The reservoir was then coated with Poly-D-Lysine and left to incubate overnight. The following day the reservoir was washed 3 times with DI water, and subsequently filled with media. Media conditions were the same as described above for HSPCs. The cell suspension was added dropwise into the reservoir. Cells were incubated at 37 °C for 3 h before taking any images. For image acquisition, the raft array was scanned on the CellRaft AIR system following system instructions and returned to a 37 °C incubator after every scan. Full array scans were conducted at various timepoints for brightfield images.

### In vivo BrdU proliferation assay

Primary transplanted mice were intraperitoneal (IP) injected with 100 mg/kg BrdU solution, 16 h post injection mice were euthanized and bone marrow cells were collected. $6 \times 10^6$ bone marrow cells were stained with stem cell flow panel as described above. Cells were then fixed and probed with a FITC BrdU antibody using BD Pharmingen BrdU Flow Kit, following manufactures protocol. Briefly, cells were fixed and permeabilized in BD cytofix/cytoperm buffer for 20 min at room temperature in the dark, followed by a wash with BD perm/wash buffer. Next, cells were incubated for 10 min on ice with BD cytoperm permeabilization buffer plus, followed by a wash with BD perm/wash buffer. The cells were then re-fixed for 5 min at room temperature for 5 min in BD cytofix/cytoperm buffer, followed by a wash with BD perm/wash buffer. Next, the cells were DNase treated to expose the incorporated BrdU. Cells were treated with 30 ug DNase/$6 \times 10^6$ cells for 1 h at 37 °C in the dark, followed by a wash with BD perm/wash buffer. Finally, the cells were stained with FITC anti-BrdU antibody and incubated for 30 min at room temperature in the dark. Cells were run on flow cytometry BD Fortessa instrument.

### Statistics and reproducibility

Statistical tests were defined for each dataset. All experiments were performed to have at least 3 biological replicates to ensure power for statistical analysis using two-sided student $t$-test. No statistical method was used to predetermine sample size. No data were excluded from the analyses. We allocated recipient mice into different groups randomly in transplant in vivo experiments. Animals in all experiment groups are sex and age matched. No other randomization was performed in the study. The investigators were not blinded to outcome assessments.

### Reporting summary

Further information on research design is available in the Nature Portfolio Reporting Summary linked to this article.

## Data availability

RNA-seq that support the findings in this study have been deposited in the Gene Expression Omnibus (GEO) database under the following accession codes, scRNA-seq GSE202421, HSC RNA-seq GSE202463, HSC/MPP HyperTRIBE RNA-seq GSE202464. Proteomic mass spectrometry data are available on the ProteomeXchange Consortium with identifier PXD019779. Source data are provided as a Source data file with this manuscript. Source data are provided with this paper.

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

## Acknowledgements

We would like to thank members of the Kharas laboratory for their discussions, helpful advice, and suggestions. L.P.V. is a Scholar of the American Society of Hematology, and acknowledges support from Damon Runyon Cancer Research Foundation, Leukemia & Lymphoma Society, K99CA229993-01 and StemCell Network for the project. F.H.C. is supported by the ASH Minority Hematology Graduate Award and by a supplement from the R01CA186702. A.K. is a Scholar of the Leukemia & Lymphoma Society and acknowledges support from NCI R01 CA204396 and R21 CA235285. H.L. is supported by K99 DK128602-01. We would also like to thank the MSKCC Integrated Genomics Operation (IGO), and molecular cytology core. M.G.K. is a Scholar of the Leukemia and Lymphoma Society and supported by the US NIH National Institute of Diabetes Digestive and Kidney Diseases Career Development Award; NIDDK NIH R01-DK101989-01A1; NCI 1R01CA193842-01, R01HL135564, R01 CA274249-01A1, and R01CA225231-01; the Kimmel Scholar Award; the V-Scholar Award; the Geoffrey Beene Award; the Starr Cancer Consortium; the Alex's Lemonade Stand A Award; the LLS Translation Research Program; the Susan and Peter Solomon Fund; and the Tri-Institutional Stem Cell Initiative 2016-014. The work is also supported by the NCI Cancer Center support grant (P30 CA08748),

## Author contributions

L.P.V. led and directed the project, designed, performed experiments, analyzed data and wrote the manuscript. F.H.C led this project, designed, performed experiments, analyzed data and wrote the manuscript. H.L.; P.C.; A.K.; and D.T.T.N supervised, analyzed experiments and provided critical suggestions. A.Pie.; E.L.Chu.; X.Y., and C.L. performed and supervised bioinformatic analysis. A.S.; E.B.; M.C.; K.Cha.; G.YQ.H.; A.J.Pin.; M.X., and L.M.K. all performed experiments. S.J. and G.C. provided critical reagents and suggestions. Y.H. provided critical reagents. M.G.K. directed the project, analyzed data and wrote the manuscript.

## Competing interests

A.K. is a consultant to Rgenta, Novartis, and Blueprint Medicines. M.G.K. is a SAB member of 858 Therapeutics and received honorarium from Kumquat, AstraZeneca and Consultancy with Transition Bio. The remaining authors declare no competing interests.

## Additional information

[1]Molecular Pharmacology Program, Memorial Sloan Kettering Cancer Center, New York, NY, USA. [2]Gerstner Sloan Kettering Graduate School of Biomedical Sciences, Memorial Sloan Kettering Cancer Center, New York, NY, USA. [3]Cold Spring Harbor Laboratory, Cold Spring Harbor, NY, USA. [4]Computational Biology Program, Memorial Sloan Kettering Cancer Center, New York, NY, USA. [5]Department of Pharmacology, Weill Cornell School of Medical Sciences, New York, NY, USA. [6]Chemical Biology Program, Memorial Sloan Kettering Cancer Center, New York, NY, USA. [7]Pharmacology Program of the Weill Cornell Graduate School of Medicine Sciences, New York, NY, USA. [8]Weill Cornell/Rockefeller/Sloan Kettering Tri-Institutional MD-PhD Program, New York, NY, USA. [9]Cell Microsystems, Inc., Durham, NC, USA. [10]Department of Biochemistry and Chemistry, La Trobe Institute for Molecular Science, La Trobe University, Melbourne, VIC 3086, Australia. [11]Centre for Haemato-Oncology, Barts Cancer Institute, Queen Mary University of London, London, UK. [12]Tow Center for Developmental Oncology, Department of Pediatrics, Memorial Sloan Kettering Cancer Center, New York, NY, USA. [13]Departments of Pediatrics, Pharmacology, and Physiology & Biophysics, Weill Medical College of Cornell University, New York, NY, USA. [14]Terry Fox Laboratory, British Columbia Cancer Research Centre, Vancouver, BC, Canada. [15]Faculty of Pharmaceutical Sciences, University of British Columbia, Vancouver, BC, Canada. [16]These authors contributed equally: Florisela Herrejon Chavez, Ly P. Vu. [17]These authors jointly supervised this work: Ly P. Vu, Michael G Kharas. ✉e-mail: lvu@bccrc.ca; kharasm@mskcc.org

