## [Peer Review File · Nature Communications]

RNA binding protein SYNCRIP maintains proteostasis and self-renewal of hematopoietic stem and progenitor cellsReviewers' comments:

Reviewer #1 (Remarks to the Author):

- What are the noteworthy results?

The manuscript from Vu et al., describes the analysis of the RNA binding protein Syncrip in hematopoiesis. Using a conditional allele they demonstrate that somatic deletion of Syncrip is well tolerated. However, once the bone marrow is transplanted they observed reduced repopulation capacity and serial transplant potential. They provide detailed analysis and FACS. Using scRNA-seq they identify a defect in a relatively newly described population of HSCs termed the aHSC. Using transcriptomics and proteomics they describe that Syncrip deficient hematopoietic cells have a defect in a number of pathways, notably the unfolded protein response and epichaperone network and they demonstrate a decrease in CDC42/cell polarity processes.

Overall, I think this is an interesting study and one which is important as it broadens our understanding of a poorly understood class of protein – RNA binding proteins. This demonstrates a role in normal hematopoiesis and in particular a role in HSC behaviour and transplant functions.

- Will the work be of significance to the field and related fields? How does it compare to the established literature? If the work is not original, please provide relevant references.
I think this study will be well received and also be of interest to both HSC biologists/experimental hematology and also RNA researchers, with the clear demonstration of an in vivo function and role for an RNA binding protein. The work is original and an extension of a theme that the lab has been working on for a number of years.

- Does the work support the conclusions and claims, or is additional evidence needed?

I think the authors need further evidence to support their conclusion.

The in vivo analysis and FACS is overall well performed (outside of my issue with the control – see comments below). I think there is insufficient evidence provided to exclude a homing or engraftment defect, particularly given the demonstrated change in polarity and CDC42 that the author provide. This pathway is associated with homing and engraftment (PMID: 34661963; PMID: 30254339; PMID: 22560076; PMID: 29891535; PMID: 17360364). The literature would most strongly indicate that the phenotypes observed could be assigned to a reduced engraftment/lodging in the BM or interaction/retention in the niche following transplant. This needs to be more completely explored before the aHSC model is fully supported as the alternative interpretation.

The hypertribe approach has been described by this group before.

I am interested in the strength of calling bound substrates when the authors state that “We mostly found one to two edit sites per transcript and few exceptions of transcripts with more than 4 edited sites.” – this is a limitation of the hypertribe approach and with one or two sites per transcript I am surprised that there is the level of confidence to analyse them in such detail. The majority of ADAR mediated A-to-I editing is less than 10% efficient and can appear largely stochastic on these substrates so the level of modification by the hypertribe I think warrants a greater degree of stringency and caution with interpretation. Many ADAR mediate sites will be in the 3'UTR and a 10% difference (as applied) would be a hard bar for most low ADAR mediated editing events to demonstrate a difference as this can be one or two reads difference. The intersection with the proteomics and discussion around this in lines 370-374 could be considered – potentially the lack of correlation is because hypertribe does not give a strong signal.

Analysis in line 360-364 – I realise that the Brown et al., 2006; Ivanova et al., 2002 gene signature have been extensively used but surely there are better signatures, more recent and from more highly purified populations from RNA-seq that can be applied to these datasets which would provide similar information.

In vivo transplant would really be needed to support the CDC42 data. I don't believe in vitro serial

colony assays are a read out of HSC activity (despite the enthusiasm with which they are used). The majority of colonies described are myeloid committed (GM, M) which don't require an HSC source to arise directly. In vivo transplant could however be considered an unreasonable request due to both resources and time required. I think more detailed analysis of the homing/engraftment would be a better revision.

- Are there any flaws in the data analysis, interpretation and conclusions? Do these prohibit publication or require revision?

I think the interpretation in the absence of homing/engraftment data is open to debate. These studies should be provided given the extensive literature linking reduced CDC42 to defects in engraftment.

Line 203 – I am intrigued that the authors report 50,000 significant genes from RNA-seq – as far as I was aware the mouse has a consensus of 20-25,000 genes (potential protein coding; no non-coding and non-polyadenylated). I can't work out where the extra genes have come from. This may just reflect how this is worded but it should be clarified.

- Is the methodology sound? Does the work meet the expected standards in your field?

I do not believe that the appropriate control has been used – a Mx-Cre-ve Syncripfl/fl mouse injected with pIpC is one control, a control that hasn't been described is an Mx-Cre+ve Syncrip+/+ or Mx-Cre+ve Syncripfl/+ (heterozygous) that has been injected with pIpC. It is crucial to include either/both of these controls to control for toxicity from Cre activation and for recombination of the DNA which can induce DNA damage response in cells. The failure to include these controls is an issue for the phenotyping post pIpC dosing. I think, that whilst ideal, these controls are not essential for the interpretation of the role of Syncrip deletion after mice have been pIpC dosed and then allowed sufficient time (>12 weeks) to recover and re-establish hematopoiesis. The secondary transplants from previous pIpC treated animals demonstrate reduced repopulating capacity and are consistent with a reduced long-term HSC functional pool.

The two dose schedule of pIpC is a newer iteration of this and I would like to see genomic PCR on a more purified fraction of cells (such as LSK) to demonstrate efficient recombination in this schedule. The recombination as shown on PCR is impressively efficient for such a short exposure acute stimulus when compared to other systems like R26-CreER and tamoxifen for deleting in HSCs.

- Is there enough detail provided in the methods for the work to be reproduced?

For the most part I think sufficient detail has been provided.

I could not determine clearly that the Neo had been deleted from the line used for experiments as there is no mention of a Flp delete strain anywhere that I can find. This should be clearly described if this is or isn't the case.

Other comments:

Line 126 – I can't work out why Subramanian is reference here (GSEA method)

Reviewer #2 (Remarks to the Author):

Vu et al. had previously identified the RNA binding protein SYNCRIP as being required to maintain myeloid leukemias but not normal hematopoiesis (Vu et al., 2017). In this study, they investigated the role of SYNCRIP in hematopoietic stem cells (HSCs). They conclude that SYNCRIP is not required for normal hematopoiesis but is required to maintain "reserve" HSCs. The authors provide evidence that in the absence of SYNCRIP, the cell cycle status of HSCs does not change but they lose self-renewal potential and exhibit pervasive defects including proteotoxic stress, altered transcriptional profile, increased protein synthesis, decreased CDC42 function, and altered tubulin staining. Despite these broad defects, they conclude that the functional deficit in SYNCRIP deficient HSCs reflects the reduction in CDC42 expression and altered cell polarity in culture. It is not clear whether the decrease

in CDC42 expression is responsible for the other defects, such as proteotoxic stress and altered transcriptional profile, or whether increased CDC42 expression rescues the function of SYNCRIP deficient HSCs in vivo. They imply that the conceptual advance is to identify a mechanism that's required by reserve HSCs but not by other stem/progenitor cells; however, the data show that SYNCRIP does regulate other stem/progenitor cells.

1. The paper is written as though SYNCRIP is preferentially required by "reserve" HSCs and not by other hematopoietic progenitors. However, the evidence is fragmentary. There is no data on the SYNCRIP expression pattern in the paper but other published data indicate that SYNCRIP is ubiquitously expressed, calling into question the suggestion that it mainly regulates reserve HSCs. The authors show that SYNCRIP deletion does not alter bone marrow or spleen cellularity, suggesting no gross hematopoietic defects, but they don't seem to provide data on blood cell counts or on the frequencies of most hematopoietic progenitors in the bone marrow. They do show altered frequencies of MPPs and reduced frequencies of colony-forming progenitors, demonstrating that the effects are not specific to reserve HSCs.
2. The most interesting and curious phenotype is that the authors report that SYNCRIP deficiency increased the rate of cell division among MPPs but not among HSCs. This further shows that the effect of SYNCRIP is not specific to reserve HSCs but it's also curious because it's very surprising that such extensive functional defects could be observed in HSCs without any effect on HSC cell cycle status.
3. The atypical HSCs are not well characterized. Has anyone seen these cells before? Have they been confirmed to have HSC activity in transplantation assays? Are atypical HSCs induced by hematopoietic stress? What is the frequency of these cells in the bone marrow? Are they dividing or not dividing? What percentage of HSCs fall within this atypical subset? The authors seem to assume that the atypical HSCs correspond to the reserve HSC population but it's not clear how strong the evidence is for this. There are some similarities in gene expression, though it's not clear how similar the gene expression profiles are.
4. The authors claim in the abstract that "Loss of SYNCRIP alters the developmental trajectory of reserve HSCs" but they offer limited evidence for this.
5. The paper is written as though SYNCRIP is preferentially regulating a biologically distinct subset of reserve HSCs but not much information is provided on these cells either. What fraction of HSCs fall within this reserve population? What's the evidence that SYNCRIP deficiency causes a defect in the reserve HSCs but not more broadly in all HSCs and MPPs? While a few labs have used the term "reserve", the appropriateness of this characterization is debatable. Some HSCs divide less frequently than other HSCs, but all HSCs appear to go into cycle regularly and to contribute to hematopoiesis. There don't seem to be any HSCs that are truly held in reserve, not contributing at all to normal hematopoiesis.
6. The authors show that SYNCRIP deficiency increases protein synthesis in MPPs. The pervasive phenotypes observed in MPPs contrasts with the suggestion that SYNCRIP preferentially regulates reserve HSCs. It seems most likely that it broadly regulates primitive hematopoietic stem/progenitor cells.
7. It is surprising that SYNCRIP deficiency increases global protein synthesis while decreasing CDC42 expression. Do the authors think this reflects unrelated functions of SYNCRIP?
8. The authors conclude that the reduction in CDC42 expression in the absence of SYNCRIP leads to defects in HSC cell polarity; however, since this was tested in isolated HSCs in culture, it's not clear whether this is relevant to the in vivo situation. Cell polarity influences cell function in the context of asymmetric cues within a tissue. It's not clear what asymmetries among isolated HSCs in culture mean.
9. The authors over-interpret their results by asserting that "cell polarity is connected with inheritance of fate determinants in daughter cells during division, including during asymmetric versus symmetric cell division". This hasn't been shown convincingly among HSCs in vivo. Studies of the asymmetric division of Numb among dividing HSCs in culture are of uncertain relevance in vivo.
10. The authors used a retroviral vector to overexpress CDC42 in sorted LSK cells from control and SYNCRIP deficient mice. Overexpression of CDC42 partially rescued the reduction in colony formation by SYNCRIP deficient stem/progenitor cells. However, the authors didn't test whether overexpression

of CDC42 could rescue the reconstituting potential of SYNCRIP deficient stem/progenitor cells in vivo.

Below we provide a point-by-point response to the reviewer's comments and how we have addressed their concerns. We thank the reviewers for their thoughtful and constructive comments.

Reviewer #1 (Remarks to the Author):

- What are the noteworthy results?

The manuscript from Vu et al., describes the analysis of the RNA binding protein Syncrip in hematopoiesis. Using a conditional allele they demonstrate that somatic deletion of Syncrip is well tolerated. However, once the bone marrow is transplanted they observed reduced repopulation capacity and serial transplant potential. They provide detailed analysis and FACS. Using scRNA-seq they identify a defect in a relatively newly described population of HSCs termed the aHSC. Using transcriptomics and proteomics they describe that Syncrip deficient hematopoietic cells have a defect in a number of pathways, notably the unfolded protein response and epichaperone network and they demonstrate a decrease in CDC42/cell polarity processes.

Overall, I think this is an interesting study and one which is important as it broadens our understanding of a poorly understood class of protein – RNA binding proteins. This demonstrates a role in normal hematopoiesis and in particular a role in HSC behaviour and transplant functions.

- Will the work be of significance to the field and related fields? How does it compare to the established literature? If the work is not original, please provide relevant references.

I think this study will be well received and also be of interest to both HSC biologists/experimental hematology and also RNA researchers, with the clear demonstration of an in vivo function and role for an RNA binding protein. The work is original and an extension of a theme that the lab has been working on for a number of years.

- Does the work support the conclusions and claims, or is additional evidence needed?

I think the authors need further evidence to support their conclusion.

The in vivo analysis and FACS is overall well performed (outside of my issue with the control – see comments below). I think there is insufficient evidence provided to exclude a homing or engraftment defect, particularly given the demonstrated change in polarity and CDC42 that the author provide. This pathway is associated with homing and engraftment (PMID: 34661963; PMID: 30254339; PMID: 22560076; PMID: 29891535; PMID: 17360364). The literature would most strongly indicate that the phenotypes observed could be assigned to a reduced engraftment/lodging in the BM or interaction/retention in the niche following transplant. This needs to be more completely explored before the aHSC model is fully supported as the alternative interpretation.

We want to point out that in our manuscript, to exclude the impact of SYNCRIP in niche interaction and evaluate the cell-intrinsic function of SYNCRIP, we performed the experiments where donor cells (*Syncrip* cKO and *Syncrip* WT) were first allowed to engraft at the comparable levels in recipient animals prior to SYNCRIP deletion (Figure 2). In this setting, we demonstrated that deletion of SYNCRIP reduced chimerism of KO cells overtime and resulted in ineffective hematopoiesis after lethal irradiation. Importantly, all molecular characterizations were also performed in this setting where any influence of engraftment or lodging was already excluded.

At the same time, we agree that a direct assessment of in a homing assay will help further support our findings. Therefore, we conducted homing experiments for the secondary transplantation and demonstrated that within 16 hours post transplantation, donor cells sorted (LSK) from both *Syncrip* WT and KO homed successfully to bone marrow of recipient animals. The data is shown below and included in the revised manuscript as new Figure S2H-I.

1st BMT - plpC → 2nd BMT
 Homing-16h post transplant

1st BMT - plpC → 2nd BMT
 homing 16h post transplant

In addition, we showed that there was only a modest reduction in chimerism in *Syncrip* KO 1 week after transplantation, strongly indicating that the loss of engraftment observed at later time points is due to loss of long-term engraftment potential of HSCs. The data is shown below and included in the revised manuscript as figure 2F (16 weeks) and new Figure 2G (1 weeks).

BM → 1st BMT - plpC → 2nd BMT
 16 weeks

2nd BMT
 1 week

The hypertribe approach has been described by this group before.

I am interested in the strength of calling bound substrates when the authors state that “We mostly found one to two edit sites per transcript and few exceptions of transcripts with more than 4 edited sites.” – this is a limitation of the hypertribe approach and with one or two sites per transcript I am surprised that there is the level of confidence to analyses them in such detail. The majority of ADAR mediated A-to-I editing is less than 10% efficient and can appear largely stochastic on these substrates so the level of modification by the hypertribe I think warrants a greater degree of stringency and caution with interpretation. Many ADAR mediate sites will be in the 3’UTR and a 10% difference (as applied) would be a hard bar for most low ADAR mediated editing events to demonstrate a difference as this can be one or two reads difference. The intersection with the proteomics and discussion around this in lines 370-374 could be considered – potentially the lack of correlation is because hypertribe does not give a strong signal.

We thank the reviewer for the comment as it allows us to clarify and further highlight the effectiveness and novelty of the HyperTRIBE assay for SYNCRIP.

In the original method development papers (McMahon et al, Cell 2016; Xu et al, RNA 2018), the Rosbach group standardized the assay to score highly confident editing sites to have at least 10% editing and a minimum of 10 and 20 reads at that site for TRIBE and HyperTRIBE respectively to remove potential noise from sequencing errors. The same threshold of 10% was applied in differential editing activity to also account for potential noise from sequencing errors which can lead to false positive hits. The method has been used and validated by our group in published studies i.e. Nguyen et al, Nature Communication 2020 and Cheng et al, Cancer Cell 2021.

In our dataset, without the overexpression of SYNCRIP-ADAR (empty vector -EV control), we observed in HSCs total 1833 edited events and 372 edited events with greater than 10% edited frequency and in MPP total 1669 edited events and 272 edited events with greater than 10% edited frequency (Freq >0.1-data shown in supplemental figure 5B). These account for about 20% and 16% ADAR mediated A-to-I editing of total editing events detected in HSCs and MPPs respectively. With the overexpression of SYNCRIP-ADAR, we identified in HSCs total edited events and 1600 edited events with greater than 10% edited frequency and in MPP total 3602 edited events and 1163 edited events with greater than 10% edited frequency (Freq>0.1-data shown in supplemental figure 5B). These editing events correspond to approximately 42% and 32% of total ADAR mediated A-to-I editing found in HSCs and MPPs respectively. Supplemental figure 5B and the pie charts below illustrate the number and percentage of high confident (Freq>0.1) editing events.

HSC EV
total =1833 events

HSC S-ADAR
total =3761 events

MPP EV
total =1669 events

MPP S-ADAR
total =3602 events

■ 0 < Freq < 0.1
■ Freq > 0.1

The data indeed demonstrated a highly efficient editing of our fusion SYNCRIP-ADAR protein as well as a strong editing signal detected by RNA-sequencing analysis. Importantly, the clear difference between the control (EV) vs. SYNCRIP-ADAR therefore allowed us to apply a stringent cutoff to select for direct targets of SYNCRIP. While we recognized that there are potential targets that could be missed by the stringent cutoff, we aimed to adhere to the standard for target discovery where the methodology has always been designed to avoid false positive results.

Analysis in line 360-364 – I realise that the Brown et al., 2006; Ivanova et al., 2002 gene signature have been extensively used but surely there are better signatures, more recent and from more highly purified populations from RNA-seq that can be applied to these datasets which would provide similar information.

Comment 3: Analysis in line 360-364 – I realize that the Brown et al., 2006; Ivanova et al., 2002 gene signature have been extensively used but surely there are better signatures, more recent and from more highly purified populations from RNA-seq that can be applied to these datasets which would provide similar information.

We have performed additional analysis to support the results. The data is showed below and included in the manuscript as supplemental figure S5L-O.

In vivo transplant would really be needed to support the CDC42 data. I don't believe in vitro serial colony assays are a read out of HSC activity (despite the enthusiasm with which they are used). The majority of colonies described are myeloid committed (GM, M) which don't required an HSC source to arise directly. In vivo transplant could however be considered an unreasonable request due to both resources and time required. I think more detailed analysis of the homing/engraftment would be a better revision.

We agree with this comment and have tested if CDC42 overexpression can rescue the defect *in vivo*. *Syncrip*^{Δ/Δ} and *Syncrip*^{f/f} LSK cells from primary mice were transduced with CDC42 overexpression (OV) retrovirus, and 1x10⁴ transduced LSK cells were transplanted into primary CD45.1 recipients. Chimerism in the bone marrow was evaluated at various timepoints. We observed that CDC42 OV did not improve SYNCRIP KO HSPCs' engraftment deficiency at the level of total bone marrow and multiple progenitors. However, OV of CDC42 resulted in preservation of long-term engraftment within the LT-HSCs (data shown below). The data suggested that while CDC42 alone is not sufficient to rescue the full reconstitution potential of SYNCRIP deficient HSPCs, CDC42 at least in part is responsible for SYNCRIP's role in maintaining reconstitution in LT-HSCs. The data also suggested that there is a context-dependent targets and function for SYNCRIP between HSCs vs. progenitors and possibly downstream hematopoietic cells. Our multi-omics analysis in fact identified many other SYNCRIP targets and pathways it influences, including other Rho GTPases and other RPBs. It is conceivable that in the context of progenitors and total bone marrow engraftment, multiple targets are required. We have included the discussion of the results in discussion section.

• Are there any flaws in the data analysis, interpretation and conclusions? Do these prohibit publication or require revision? I think the interpretation in the absence of homing/engraftment data is open to debate. These studies should be provided given the extensive literature linking reduced CDC42 to defects in engraftment.

We have now provided extensive homing data.

Line 203 – I am intrigued that the authors report 50,000 significant genes from RNA-seq – as far as I was aware the mouse has a consensus of 20-25,000 genes (potential protein coding; no non-coding and non-polyadenylated). I cant work out where the extra genes have come from. This may just reflect how this is worded but it should be clarified.

We thank the reviewer for pointing out the typos in our manuscript so that we can clarify the description. The number “50,000” in line 203 was now corrected to be the number of total cells, which is 49,599 cells included in our scRNA-seq analysis (supplemental table 1- cells with cut off for gene Mt<0.15; log count >3.1). As also indicated in supplemental table 1, after removing mitochondrial and ribosomal genes and applying cutoff for genes identified in more than 3 cells, we included 18,011 genes in our analysis.

• Is the methodology sound? Does the work meet the expected standards in your field?

I do not believe that the appropriate control has been used – a Mx-Cre-ve Syncripfl/fl mouse injected with plpC is one control, a control that hasn't been described is an Mx-Cre+ve Syncrip+/+ or Mx-Cre+ve Syncripfl/+ (heterozygous) that has been injected with plpC. It is crucial to include either/both of these controls to control for toxicity from Cre activation and for recombination of the DNA which can induce DNA damage response in cells. The failure to include these controls is an issue for the phenotyping post plpC dosing. I think, that whilst ideal, these controls are not essential for the interpretation of the role of Syncrip deletion after mice have been plpC dosed and then allowed sufficient time (>12 weeks) to recover and re-establish hematopoiesis. The secondary transplants from previous plpC treated animals demonstrate reduced repopulating capacity and are consistent with a reduced long-term HSC functional pool.

Comment 5: I do not believe that the appropriate control has been used – a Mx-Cre-ve Syncripfl/fl mouse injected with plpC is one control, a control that hasn't been described is an Mx-Cre+ve Syncrip+/+ or Mx-Cre+ve Syncripfl/+ (heterozygous) that has been injected with plpC. It is crucial to include either/both of these controls to control for toxicity from Cre activation and for recombination of the DNA which can induce DNA damage response in cells. The failure to include these controls is an issue for the phenotyping post plpC dosing. I think, that whilst ideal, these controls are not essential for the interpretation of the role of Syncrip deletion after mice have been plpC dosed and then allowed sufficient time (>12 weeks) to recover and re-establish hematopoiesis. The secondary transplants from previous plpC treated animals demonstrate reduced repopulating capacity and are consistent with a reduced long-term HSC functional pool.

We thank the reviewer for the comments. We agree with these concerns and have conducted primary transplants with SYNCRIP heterozygous bone marrow cells to ensure that our phenotype is not a result of Cre toxicity. In the non-competitive transplant for these heterozygous cells, we see no change in engraftment. This indicates that the engraftment defects we see in our homozygous transplants are not a result of Cre expression or toxicity. However, we do see a reduction in engraftment when heterozygous SYNCRIP KD cells are transplanted in a competitive setting, this suggests that even with only half of SYNCRIP depleted these cells already lose some of their ability to compete with WT HSCs. The data is shown below and has now been included in **new supplemental Figure S1S-U** in the manuscript to further support our conclusion on the role of SYNCRIP in maintaining the self-renewal potential of HSCs.

The two dose schedule of plpC is a newer iteration of this and I would like to see genomic PCR on a more purified fraction of cells (such as LSK) to demonstrate efficient recombination in this schedule. The recombination as shown on PCR is impressively efficient for such a short exposure acute stimulus when compared to other systems like R26-CreER and tamoxifen for deleting in HSCs.

We thank the reviewer for the comments and to recognize our system as “impressively efficient”. In fact, the two-dose schedule of plpC has actually been used effectively by our group in several other published studies and by many other groups in the field including:

Arginine methyltransferase PRMT5 is essential for sustaining normal adult hematopoiesis.

Liu F, Cheng G, Hamard PJ, Greenblatt S, Wang L, Man N, Perna F, Xu H, Tadi M, Luciani L, Nimer SD. *J Clin Invest*. 2015 Sep;125(9):3532-44. doi: 10.1172/JCI81749. Epub 2015 Aug 10. PMID: 26258414

Jak1 Integrates Cytokine Sensing to Regulate Hematopoietic Stem Cell Function and Stress Hematopoiesis.

Kleppe M, Spitzer MH, Li S, Hill CE, Dong L, Papalexi E, De Groote S, Bowman RL, Keller M, Koppikar P, Rapaport FT, Teruya-Feldstein J, Gandara J, Mason CE, Nolan GP, Levine RL. *Cell Stem Cell*. 2018 Feb 1;22(2):277. doi: 10.1016/j.stem.2017.12.018. PMID: 29395057

m6A RNA Methylation Maintains Hematopoietic Stem Cell Identity and Symmetric Commitment.

Cheng Y, Luo H, Izzo F, Pickering BF, Nguyen D, Myers R, Schurer A, Gourkanti S, Brüning JC, Vu LP, Jaffrey SR, Landau DA, Kharas MG. *Cell Rep*. 2019 Aug 13;28(7):1703-1716.e6. doi: 10.1016/j.celrep.2019.07.032. PMID: 31412241

HyperTRIBE uncovers increased MUSASHI-2 RNA binding activity and differential regulation in leukemic stem cells. Nguyen DTT, Lu Y, Chu KL, Yang X, Park SM, Choo ZN, Chin CR, Prieto C, Schurer A, Barin E, Savino AM, Gourkanti S, Patel P, Vu LP, Leslie CS, Kharas MG.

Nat Commun. 2020 Apr 24;11(1):2026. doi: 10.1038/s41467-020-15814-8. PMID: 32332729

Transcriptional control of CBX5 by the RNA binding proteins RBMX and RBMXL1 maintains chromatin state in myeloid leukemia. Prieto C, Nguyen DTT, Liu Z, Wheat J, Perez A, Gourkanti S, Chou T, Barin E, Velleca A, Rohwetter T, Chow A, Taggart J, Savino AM, Hoskova K, Dhodapkar M, Schurer A, Barlowe TS, Vu LP, Leslie C, Steidl U, Rabadan R, Kharas MG.

We believe that the extensive literature represents strong evidence for the efficiency of the method. In addition, in the study, we have assessed the efficiency of SYNCRIP depletion not only by immunoblots of total bone marrow cells but also by a more stringent evaluation by immunofluorescent (IF) of sorted HSCs and MPPs cells. The assessment of presence/absence of functional protein is the gold standard to evaluate biological consequences of loss of function. Representative IF and quantification of IF signals were shown in the manuscript as figure 6A-B. Additional representative IFs are shown below together with the quantitative summary data (figure 6B).

• Is there enough detail provided in the methods for the work to be reproduced?

For the most part I think sufficient detail has been provided.

I could not determine clearly that the Neo had been deleted from the line used for experiments as there is no mention of a Flp delete strain anywhere that I can find. This should be clearly described if this is or isn't the case.

We have now included the below scheme to clarify the design and generation of our novel genetic model in new supplemental figure S1B. A Neo allele was included for selection and flipped out prior to generation of founder animals, which was then crossed with Mx-1 Cre.

Other comments:

Line 126 – I cant work out why Subramanian is reference here (GSEA method)

This was the original reference for the GSEA method.

Reviewer #2 (Remarks to the Author):

Vu et al. had previously identified the RNA binding protein SYNCRIP as being required to maintain myeloid leukemias but not normal hematopoiesis (Vu et al., 2017). In this study, they investigated the role of SYNCRIP in hematopoietic stem cells (HSCs). They conclude that SYNCRIP is not required for normal hematopoiesis but is required to maintain “reserve” HSCs. The authors provide evidence that in the absence of SYNCRIP, the cell cycle status of HSCs does not change but they lose self-renewal potential and exhibit pervasive defects including proteotoxic stress, altered transcriptional profile, increased protein synthesis, decreased CDC42 function, and altered tubulin staining. Despite these broad defects, they conclude that the functional deficit in SYNCRIP deficient HSCs reflects the reduction in CDC42 expression and altered cell polarity in culture. It is not clear whether the decrease in CDC42 expression is responsible for the other defects, such as proteotoxic stress and altered transcriptional profile, or whether increased CDC42 expression rescues the function of SYNCRIP deficient HSCs in vivo. They imply that the conceptual advance is to identify a mechanism that’s required by reserve HSCs but not by other stem/progenitor cells; however, the data show that SYNCRIP does regulate other stem/progenitor cells.

Please see pt by pt response for the comments raised here..

1. The paper is written as though SYNCRIP is preferentially required by “reserve” HSCs and not by other hematopoietic progenitors. However, the evidence is fragmentary. There is no data on the SYNCRIP expression pattern in the paper but other published data indicate that SYNCRIP is ubiquitously expressed, calling into question the suggestion that it mainly regulates reserve HSCs. The authors show that SYNCRIP deletion does not alter bone marrow or spleen cellularity, suggesting no gross hematopoietic defects, but they don’t seem to provide data on blood cell counts or on the frequencies of most hematopoietic progenitors in the bone marrow. They do show altered frequencies of MPPs and reduced frequencies of colony-forming progenitors, demonstrating that the effects are not specific to reserve HSCs.

We thank the review for the comments. As the reviewer noted, we did not observe any gross defects and overt changes in frequencies of both progenitors and stem cell compartments during static hematopoiesis in primary animals upon SYNCRIP deletion. We also followed blood count of SYNCRIP KO animals up to 22 weeks post deletion and observed reduced lymphocyte count, resulting in reduced total white blood count (WBC), a slight reduction in total platelet count (PLT count) but no significant change in red blood count (RBC) and hemoglobin (HGB) level. These data were shown below and now included data in the manuscript as **new supplemental figure S1C-G**. These data further support our conclusion on the minimal requirement for SYNCRIP function for the majority of cells within the hematopoietic systems.

As shown below and now included in the manuscript in **new supplemental figure 1A**, we observed that SYNCRIP is highly expressed in the stem and progenitor cell compartments.

In our manuscript, we showed that there was a reduction in chimerism in 1st transplantation starting at 8 weeks post engraftment. While long term engraftment is mainly driven by HSCs, multiple progenitors MPPs also contribute to engraftment at early timepoints. Therefore, we wrote in our manuscript *“These data indicate that SyncrIP^{Δ/Δ} HSPCs exhibit reduced repopulating potential compared to the control cells”* to indicate that the effect can be attributed to deficiency in both progenitor and stem cell compartments. We appreciate the reviewer’s comment and have added the sentence to better describe the results *“The early reduction in engraftment can at least in part be attributed to the reduced engraftment capability of progenitors”*.

At the same time, we want to point out that expression of RNA binding proteins (RBPs) does not necessarily translate to function and there are several precedents that RBP's function is highly context dependent. In fact, the most drastic effect we observed is the loss of engraftment in a competitive transplantation at both early and late time points as well as in secondary transplantation where MPPs activity is no longer relevant. In addition, the scRNA-sequencing data revealed that the unfolded protein response was specifically deregulated in the HSC compartments but not the MPPs (Figure 3 K, L vs. new supplemental figure 3K). Given these observations and the lack of understanding of how HSC self-renewal is regulated under stress and its exquisite dependency on protein quality control (Hidalgo San Jose et al., 2020), we focused our study on dissecting the role of SYNCRIP in HSCs. We also believe that it is the area where our data can bring the most insights and advance in the field.

2. The most interesting and curious phenotype is that the authors report that SYNCRIP deficiency increased the rate of cell division among MPPs but not among HSCs. This further shows that the effect of SYNCRIP is not specific to reserve HSCs but it's also curious because it's very surprising that such extensive functional defects could be observed in HSCs without any effect on HSC cell cycle status.

We thank the review for the comment and interest in our findings. We agreed that the effects of SYNCRIP loss is not only observed in HSCs but also MPPs and indeed, we wrote in the manuscript "*The data suggest that the early drop in donor engraftment after Syncrip deletion and loss of regenerative potential was associated with enhanced cycling in the MPPs*". The data supported our interpretation on the reduction in engraftment at early time point as we discussed in comment#1. However, the changes in MPPs were not sufficient to explain the phenotypes we observed in long-term and secondary engraftment, which is mainly driven by HSCs' activity.

We previously used pyronin Y and Hoechst staining flow cytometry analysis to evaluate cell cycle status of HSPCs. We observed a trend of changes in cell cycle in HSCs at 3 weeks post deletion, the replicated data does not reach significance. To further address the reviewer's comment on potential effects of SYNCRIP deletion on cell cycle and cell proliferation, we performed additional sets of new experiments to definitively examine SYNCRIP's influence on these cellular processes.

First, we performed BrdU labeling and tracing in vivo to determine whether SYNCRIP loss enhances cycling and induce exit of quiescent HSCs. We found that there was no significant change in percentage of BrdU positive cells across all HSPC compartments in vivo, except for a modest increase in the MPP2. The data shown below is included in the revised manuscript as new supplemental figure S2J.

To directly assess cell division in HSCs, we plated single-cells from WT and KO mice into individual well and tracked their division by imaging using the CellRaft AIR System (Cell Microsystems). Over a period of 60 hours, we visualized cell division and calculated how long it took our HSCs to start dividing in vitro. We found that while there was a slight increase in the mean value of time to first cell division (12.23 hours in *Syncrip*^{Δ/Δ} vs. 10.16 hours in *Syncrip*^{f/f}), the cumulative outputs were no difference between the two conditions, suggesting that the impact of SYNCRIP loss is modest. The data is shown below and included in the manuscript as new supplemental figure S2K and new figure 2K.

To further evaluate the impact of SYNCRIP deletion on HSPC proliferation, we performed CFSE labeling experiments where retention of CFSE signal correlates with less division while the loss of CFSE in cells indicates dilution of CFSE upon cell division. We tracked CFSE signals in HSPCs upon engraftment in recipients after 1 weeks of transplantation. We observed that there was no difference in CFSE patterns in KO vs. WT in both LSK and HSC populations. The data is shown

below and included in the revised manuscript as **new supplemental figure S2L-M and new figure 2L-M.**

Overall, these new data strongly indicate that SYNCRIP plays a negligible role in controlling cell cycle and proliferation of HSCs. The results strongly support that the unique dependence of HSCs on SYNCRIP is to maintain a balanced proteome to sustain self-renewal capacity is indeed a major role for SYNCRIP in HSCs.

3. The atypical HSCs are not well characterized. Has anyone seen these cells before? Have they been confirmed to have HSC activity in transplantation assays? Are atypical HSCs induced by hematopoietic stress? What is the frequency of these cells in the bone marrow? Are they dividing or not dividing? What percentage of HSCs fall within this atypical subset? The authors seem to assume that the atypical HSCs correspond to the reserve HSC population but it's not clear how strong the evidence is for this. There are some similarities in gene expression, though it's not clear how similar the gene expression profiles are.

We thank the reviewer for raising this important point. While there are available dataset of scRNAseq of ckit enriched mouse bone marrow cells, the majority of datasets were performed using primary animals. In our study, we performed the analysis using transplanted cells which underwent plpC treatment. These cells therefore prior to examination were subjected to regenerative stress. The response of “reserve” HSCs to stress signal allowed us to identify the atypical HSC (aHSC) population, which is not previously characterized in naïve bone marrow cells. Our scRNAseq analysis (n=3/each condition and n=6 total) identified aHSC among total analyzed ckit enriched cells at the average frequency of ~4% and the HSC population also is ~4% average frequency. Due to the lack of surface markers to isolate the aHSC, we are not able to separately evaluate HSC vs. aHSC in functional assays.

While we report the presence of the aHSCs, our geneset enrichment analysis (GSEA) strongly indicated that the aHSCs resemble the low input HSCs described by Rodriguez-Fraticelli et al., 2020. It is noted that while the low input HSC population was functionally characterized by lineage tracing experiments, there is also no surface marker to isolate and solely evaluate its function and the identification of the population is based on gene expression profile. To further support our observation and identification of aHSCs, we ran GSEA of our aHSCs against all gene lists included in Rodriguez-Fraticelli et al., 2020 for characterization of low input HSCs and found a strong enrichment of aHSC (vs. HSC) transcriptome for all gene signatures corresponding to populations with increased “self-renewal” and “reserved” HSC compartment including Cabezas-Wallsheid_HSC vs MPP4 Signature; Peitras_HSC Signature; Single Cell Signature Cabezas-Wallsheid_dormantHSC vs activatedHSC; Giladi_StemScore Signature; HSC serial transplant signature and HSC1-4 Cluster Signatures. Thus, based on these five independent datasets it is clear that the aHSCs strongly enrich for previously determined dormant and reserve HSCs. The data shown below has also now been included in the manuscript as new supplemental figure S3F-J.

Rodriguez-Fraticelli, A.E., Weinreb, C., Wang, S.W., Migueles, R.P., Jankovic, M., Usart, M., Klein, A.M., Lowell, S., and Camargo, F.D. (2020). Single-cell lineage tracing unveils a role for TCF15 in haematopoiesis. *Nature* 583, 585-589.

Cabezas-Wallscheid, N., Klimmeck, D., Hansson, J., Lipka, D.B., Reyes, A., Wang, Q., Weichenhan, D., Lier, A., von Paleske, L., Renders, S., *et al.* (2014). Identification of Regulatory Networks in HSCs and Their Immediate Progeny via Integrated Proteome, Transcriptome, and DNA Methylation Analysis. *Cell Stem Cell* *15*, 507-522.

Wilson, N.K., Kent, D.G., Buettner, F., Shehata, M., Macaulay, I.C., Calero-Nieto, F.J., Sanchez Castillo, M., Oedekoven, C.A., Evangelia, D., Schulte, R., *et al.* (2015). Combined Single-Cell Functional and Gene Expression Analysis Resolves Heterogeneity within Stem Cell Populations. *Cell Stem Cell* *16*, 712-724.

Giladi, A., Paul, F., Herzog, Y., Lubling, Y., Weiner, A., Yofe, I., Jaitin, D., Cabezas-Wallscheid, N., Dress, R., Ginhoux, F., *et al.* (2018). Single-cell characterization of haematopoietic progenitors and their trajectories in homeostasis and perturbed haematopoiesis. *Nat Cell Biol* *20*, 836-846.

Lauridsen, F.K., Lyholm Jensen, T., Rapin, N., Aslan, D., Wilhelmson, A.S., Pundhir, S., Rehn, M., Paul, F., Giladi, A., Hasemann, M.S., *et al.* (2018). Differences in Cell Cycle Status Underlie Transcriptional Heterogeneity in the HSC Compartment. *Cell Reports* *24*, 766-780.

4. The authors claim in the abstract that “Loss of SYNCRIP alters the developmental trajectory of reserve HSCs” but they offer limited evidence for this.

We thank the reviewer to pointing this out. Partition-based graph abstraction (PAGA) analysis of scRNAseq data demonstrated a connection of state between aHSC to HSC and from HSC to MPP1 followed by MPP2 (Figure 3G). Upon SYNCRIP deletion, we found that the predicted trajectory from aHSC to HSC was lost. Since the finding was based on predicted trajectory and not yet proven functionally, we now removed the statement in the abstract and included the exact description of the analysis in our result section.

5. The paper is written as though SYNCRIP is preferentially regulating a biologically distinct subset of reserve HSCs but not much information is provided on these cells either. What fraction of HSCs fall within this reserve population? What's the evidence that SYNCRIP deficiency causes a defect in the reserve HSCs but not more broadly in all HSCs and MPPs? While a few labs have used the term “reserve”, the appropriateness of this characterization is debatable. Some HSCs divide less frequently than other HSCs, but all HSCs appear to go into cycle regularly and to contribute to hematopoiesis. There don't seem to be any HSCs that are truly held in reserve, not contributing at all to normal hematopoiesis.

We thank the reviewer for the comment. As we addressed in previous comments, we characterized the consequences of SYNCRIP loss in the whole hematopoietic system (response to comment #1) and found a most significant defect in phenotypes pertained to HSCs' function. While there are sections in the manuscript highlighting the role of SYNCRIP in reserve aHSCs, we described the impact of SYNCRIP in both aHSC and HSC in our scRNA seq analysis. The altered unfolded protein responses were also observed in both populations. In addition, all follow up experiments were done using sorted HSCs based on traditional surface markers which captured all HSCs. As there is connection to the previously described low input HSCs, which are responsible for long term regeneration in secondary transplantation setting, we inferred that the activity loss observed in the secondary engraftment was largely driven by the “reserve” HSC compartment. At the same time, we recognized that the identification of the low input HSCs population is fairly new and subjected for additional validation. We have now revised the

manuscript to scale back the emphasis on reserve HSCs and to more clearly describe the impact of SYNCRIP in HSC function during regenerative stress and reduced self-renewal.

6. The authors show that SYNCRIP deficiency increases protein synthesis in MPPs. The pervasive phenotypes observed in MPPs contrasts with the suggestion that SYNCRIP preferentially regulates reserve HSCs. It seems most likely that it broadly regulates primitive hematopoietic stem/progenitor cells.

We thank the reviewer to bringing up the point that allows us to further elaborate on the differential requirement for a balanced proteostasis in HSCs vs. MPPs, which is responsible for the exquisite requirement for SYNCRIP function in HSCs. We agree with the reviewer that the impact of SYNCRIP loss on protein synthesis is broadly applicable. In fact, we also observed the phenotype in leukemia cells (Vu et al. 2017). However, the deregulated unfolded protein response was only observed in HSCs but not MPPs as demonstrated in our scRNAseq analysis, flow analysis assessments of unfolded protein and epichaperome. Therefore, while the changes in protein synthesis not specific for HSCs, the data clearly showed that the responses and dependencies of HSCs vs. MPPs to these factors are different. In fact, it has been demonstrated by LH San Jose et al. 2020 that HSCs are more vulnerable to reduced proteome quality while MPPs have the ability to dilute, hence overcome stress induced by misfolded and unfolded proteins

7. It is surprising that SYNCRIP deficiency increases global protein synthesis while decreasing CDC42 expression. Do the authors think this reflects unrelated functions of SYNCRIP?

We thank the reviewer for the comment. It has been increasingly recognized that RBPs function in diverse ways and in a context dependent manner. How an RBP function is depending on specific targets as well as its interaction with other RBPs and proteins in the cells. While the mechanisms for translation regulation by SYNCRIP is not fully understood, this study and our previous work (Vu et al. 2017) suggested that SYNCRIP globally suppresses protein synthesis while promotes translation of specific mRNA transcripts. We previously published that SYNCRIP depletion increases global protein synthesis in acute myeloid leukemia cells (AML) while also decreasing expression of its direct target HOXA9. We also shown that SYNCRIP interacts with other RBPs, possibly forming different functional complexes depending on the context and targets. In HSCs, we observed that deletion of SYNCRIP also led to increased protein synthesis. We noticed that loss of SYNCRIP influenced expression of several RNA binding proteins as well as proteins involved translational control, suggesting that the phenotype on global translation activity could be a broad secondary effect. On the other hand, we established a direct functional connection between SYNCRIP and CDC42 protein expression and validated that SYNCRIP deletion resulted in loss of CDC42 expression. These data indicated that as an RBP, SYNCRIP could play a diverse role in mediating gene expression control, hence influencing cell physiological states by different mechanisms.

8. The authors conclude that the reduction in CDC42 expression in the absence of SYNCRIP leads to defects in HSC cell polarity; however, since this was tested in isolated HSCs in culture, it's not clear whether this is relevant to the in vivo situation. Cell polarity influences cell function in the context of asymmetric cues within a tissue. It's not clear what asymmetries among isolated HSCs in culture mean.

We thank the reviewer for the comment. While we agree that environmental cues play important roles influencing cell polarity, there are intrinsic factors that determine cell polarity and division.

While it is ideal to evaluate all biological events *in vivo*, technical challenges represent significant barrier for such assessment *in vivo*. Seminal studies in the field which led to major advances in our understanding of HSC cell polarity and asymmetric vs. symmetric division were all based on *in vitro* assessments. However, while the evaluation of symmetric vs. asymmetric division was performed in an *in vitro* setting, assessment of HSC's function was examined by single cell micromanipulation and single cell transplantation into recipient animals. In several studies (Ito K et al. 2012, Yamamoto R et al. 2013, Ito K et al. 2021), two daughter cells in the paired daughter cell assay were isolated, transplanted into mice and followed for their ability to reconstitute in primary and secondary recipients. Isolated cells were then retrospectively defined as long-term, intermediate-term or short-term HSCs. The evidence presented by these studies listed below strongly supported the functional relevance *in vivo* of HSC polarity and asymmetric cell division.

Below are several seminal studies that solidly established the concept of HSC polarity and asymmetric vs. symmetric cell division:

Imaging hematopoietic precursor division in real time. Wu M, Kwon HY, Rattis F, Blum J, Zhao C, Ashkenazi R, Jackson TL, Gaiano N, Oliver T, Reya T. *Cell Stem Cell*. 2007 Nov;1(5):541-54. doi: 10.1016/j.stem.2007.08.009. PMID: 18345353

Asymmetric segregation and self-renewal of hematopoietic stem and progenitor cells with endocytic Ap2a2. Ting SB, Deneault E, Hope K, Cellot S, Chagraoui J, Mayotte N, Dorn JF, Laverdure JP, Harvey M, Hawkins ED, Russell SM, Maddox PS, Iscove NN, Sauvageau G. *Blood*. 2012 Mar 15;119(11):2510-22. doi: 10.1182/blood-2011-11-393272. Epub 2011 Dec 14. PMID: 22174158

A PML-PPAR- δ pathway for fatty acid oxidation regulates hematopoietic stem cell maintenance. Ito K, Carracedo A, Weiss D, Arai F, Ala U, Avigan DE, Schafer ZT, Evans RM, Suda T, Lee CH, Pandolfi PP. *Nat Med*. 2012 Sep;18(9):1350-8. doi: 10.1038/nm.2882. PMID: 22902876

Clonal analysis unveils self-renewing lineage-restricted progenitors generated directly from hematopoietic stem cells. Yamamoto R, Morita Y, Oebara J, Hamanaka S, Onodera M, Rudolph KL, Ema H, Nakauchi H. *Cell*. 2013 Aug 29;154(5):1112-1126. doi: 10.1016/j.cell.2013.08.007. PMID: 23993099

Lis1 regulates asymmetric division in hematopoietic stem cells and in leukemia
Bryan Zimdahl, Takahiro Ito, Allen Blevins, Jeevisha Bajaj, Takaaki Konuma, Joi Weeks, Claire S. Koechlein, Hyog Young Kwon, Omead Arami, David Rizzieri, H. Elizabeth Broome, Charles Chuah, Vivian G. Oehler, Roman Sasik, Gary Hardiman, Tannishtha Reya. *Nat Genet*. 2014 Mar; 46(3): 245-252. Published online 2014 Feb 2. doi: 10.1038/ng.2889 PMID: PMC4267534

Asymmetric lysosome inheritance predicts activation of haematopoietic stem cells.
Loeffler D, Wehling A, Schneiter F, Zhang Y, Müller-Böttcher N, Hoppe PS, Hilsenbeck O, Kokkaliaris KD, Ende M, Schroeder T. *Nature*. 2019 Sep;573(7774):426-429. doi: 10.1038/s41586-019-1531-6. Epub 2019 Sep 4. PMID: 31485073

Self-renewal of a purified Tie2+ hematopoietic stem cell population relies on mitochondrial clearance. Ito K, Turcotte R, Cui J, Zimmerman SE, Pinho S, Mizoguchi T, Arai F, Runnels JM, Alt C, Teruya-Feldstein J, Mar JC, Singh R, Suda T, Lin CP, Frenette PS, Ito K. *Science*. 2016 Dec 2;354(6316):1156-1160. doi: 10.1126/science.aaf5530. Epub 2016 Oct 13. PMID: 27738012

9. The authors over-interpret their results by asserting that “cell polarity is connected with inheritance of fate determinants in daughter cells during division, including during asymmetric versus symmetric cell division”. This hasn’t been shown convincingly among HSCs *in vivo*. Studies of the asymmetric division of Numb among dividing HSCs in culture are of uncertain relevance *in vivo*.

We thank the reviewer for the comment. However, as we addressed the comment #8, seminal studies in the field which led to major advances in our understanding of HSC cell polarity and asymmetric vs. symmetric division were all evaluated *in vitro* as in our assays. The observations made *in vitro* were then coupled and supported by functional assessments both *in vitro* and *in vivo*. Moreover, other studies and ours characterized not only NUMB but also ACTIN, TUBULIN and LAMP1 to evaluate symmetric vs. asymmetric division and how the patterns of division influence fates of daughter cells. The validity of HSC polarity and asymmetric versus symmetric cell division is supported in the HSC biology field.

In our study, we found that deregulation of cell polarity upon SYNCRIP loss is coupled with defects in partition of LAMP1-marked lysosome. To further support our inference that lysosomal defects hence influence the ability of HSCs to clear out misfolded proteins leading to disruption of proteostasis, we performed immunofluorescence imaging using tetraphenylethene maleimide (TMI) and validated our observation with flow cytometry analysis that TMI-bound unfolded proteins are accumulated in *Syncrip* KO HSCs and TMI level is correlated with LAMP1 and NUMB abundance. The data is shown below and included in the revised manuscript as **new supplemental figure S6A-C**.

We performed IF staining of TMI, LAMP1 and NUMB in a paired daughter assay and also observed that there was reduced asymmetric division of TMI. Data shown below and included as **new figure 6M-N**.

As it was previously demonstrated that HSCs, unlike MPPs cannot dilute misfolded proteins via high rate of proliferation (Hildago et al., 2019) and HSCs can deposit LAMP1-marked lysosome asymmetrically to differentiating daughter cells (Loeffler et al., 2019), we checked the pattern of co-segregation of TMI and LAMP1 in asymmetric dividing cells. Interestingly, we saw that TMI and LAMP1 is always co-segregated in asymmetric dividing cells (daughter cells with high LAMP1 have high TMI and daughter cells with low LAMP1 have low TMI). Upon SYNCRIP depletion, over 30% of paired daughter cells showed abnormal pattern of LAMP1 and TMI division. No significant change in total level of TMI was observed. Data shown below and included as new figure 6O-P and new supplemental figure S6F.

In addition, we observed that when HSCs divide symmetrically based on LAMP1 staining, regardless of LAMP1 status, TMI levels were elevated in *Syncrip* deficient HSCs. Data shown below and included as new supplemental figure S6E and new figure 6Q-R.

Taken together, the new set of data strongly supported our findings that loss of SYNCRIP resulted in defective lysosomal-mediated degradation and partition of misfolded proteins. This in turn may affect the ability of HSCs to dump out misfolded proteins to progenitor cells during asymmetric division.

10. The authors used a retroviral vector to overexpress CDC42 in sorted LSK cells from control and SYNCRIP deficient mice. Overexpression of CDC42 partially rescued the reduction in colony formation by SYNCRIP deficient stem/progenitor cells. However, the authors didn't test whether overexpression of CDC42 could rescue the reconstituting potential of SYNCRIP deficient stem/progenitor cells *in vivo*.

We agree with this comment and have tested if CDC42 overexpression can rescue the defect *in vivo*. *Syncrip*^{Δ/Δ} and *Syncrip*^{f/f} LSK cells from primary mice were transduced with CDC42 overexpression (OV) retrovirus, and 1x10⁴ transduced LSK cells were transplanted into primary CD45.1 recipients. Chimerism in the bone marrow was evaluated at various timepoints. We observed that CDC42 OV did not improve SYNCRIP KO HSPCs' engraftment deficiency at the level of total bone marrow and multiple progenitors. However, OV of CDC42 resulted in preservation of long-term engraftment within the LT-HSCs (data shown below). The data suggested that while CDC42 alone is not sufficient to rescue the full reconstitution potential of SYNCRIP deficient HSPCs, CDC42 at least in part is responsible for SYNCRIP's role in maintaining reconstitution in LT-HSCs. The data also suggested that there is a context-dependent targets and function for SYNCRIP between HSCs vs. progenitors and possibly downstream hematopoietic cells. Our multi-omics analysis in fact identified many other SYNCRIP targets and pathways it influences, including other Rho GTPases and other RPBs. It is conceivable that in the context of progenitors and total bone marrow engraftment, multiple targets are required. We have included the discussion of the results in discussion section.

CD45.2 Chimerism 8w BMA

CD45.2 Chimerism 24w BM

Reviewers' comments:

Reviewer #1 (Remarks to the Author):

The revised manuscript deals with many of the comments raised on the initial reviews. The additional data and tempering of the strength of conclusion improve the manuscript.

Specific comments:

line 156-158. The authors should reword the text as I do not believe this is the intended meaning of the comments that were part of the initial review - Not "bias from the Cre locus" - the control is needed for toxicity from Cre expression and activity (DNA damage etc) in addition to a control for the pIpC treatment and IFN response. The data should be shown as a supplemental file as it is an important control for readers to assess.

Regarding Hypertribe (in comments to reviewers) - whilst unreasonable to change or complete more experiments, the authors should consider a deaminase dead ADAR fusion rather than EV as a control. I am not sure comparison to an EV is particularly useful.

CDC42 over-expression data (in comments to reviewers) - I would encourage that this data be included as supplemental data. It is very important to acknowledge the limitations and uncertainty of the findings as well as the clear results so this would be important to include as it suggests more complexity than implied by the direct model proposed in Figure 7 (I don't see this as a flaw - I think it is better to accept that there is grey zones in results and acknowledge these). Related to this figure the WT data is hard to see due to the thickness of the line on the columns being the same colour as the individual sample dots.

New data in Supp Fig S1S-U - this conclusion could have been even stronger with a Mx-Cre+ Wt control.....

Re pIpC and genomic deletion - the authors have made an attempt to answer this. The references provided are nearly all from the lab that generated the current study so the statement of "many groups" is open to interpretation. I can accept this point and move on. However, the data that is shown (Fig 6A-B) requires clarification - how am I meant to interpret this data. What's the baseline and what is null in terms of signal? If I am assuming no protein is a score of 0 (this may be naive but the data is open to this interpretation) then the average KO cell has ~50% protein signal of a control cell (protein intensity of 1)?? I do not find this a clear answer to the question but it may be I am not interpreting it as the authors intended. The immunoblots are more easily understood.

Reviewer #2 (Remarks to the Author):

The authors have improved their paper by softening their conclusions; however, the data remain over-interpreted in multiple ways.

1. One of the key issues during the last round of review was that the authors were arguing that SYNCRIP was preferentially required by "reserve" hematopoietic stem cells (HSCs) even though the data showed that SYNCRIP is ubiquitously expressed by hematopoietic cells and functionally necessary in many different stem and progenitor cell populations, such that its deletion broadly reduces blood cell counts, even under steady state conditions. In the current manuscript they have softened this claim, but it is still made. For example the abstract starts "Tissue homeostasis is maintained after stress by engaging a subset of dormant and highly self-renewing stem cell populations". In reality, stress activates all HSCs, not just a subset of dormant stem cells. They go on to identify a previously uncharacterized "atypical HSC" population that they concluded is "specifically affected" by SYNCRIP deletion. However, there is no functional evidence that the "aHSCs" that were identified based on gene

expression profiling are actually HSCs. They could be MPPs or early restricted progenitors, which are also affected by SYNCRIP deficiency. It is not clear what these aHSCs are and the data definitely don't support the idea that they are specifically affected by SYNCRIP deficiency. Thus, the authors continue to ignore the obvious conclusion that SYNCRIP is broadly required by stem and progenitor cells in favor of the flashier idea that there is a novel subset of dormant HSCs that is preferentially affected. The authors have neither shown that the atypical HSCs are actually HSCs, nor that they are preferentially affected by SYNCRIP deficiency.

2. The authors reasoning that they "inferred that the activity loss observed in the secondary engraftment was largely driven by the "reserve" HSC compartment is not sound. When defects in stem/progenitor cells lead to reduced blood cell counts, there is a broad activation of HSCs and hematopoiesis in an effort to restore normal blood cell counts. This broadly depletes the self-renewal potential of HSCs, reducing their reconstituting potential, particularly upon serial transplantation. This phenotype has been observed in scores of mutant mice and never shown to be driven only by effects on "reserve" HSCs.

3. The authors have also softened their claims about CDC42, acknowledging now that it gives only a partial rescue; however, I remain uncomfortable with their claims about effects of SYNCRIP on "cell polarity" and asymmetric division for the reasons described in my original review. It is not clear whether the phenotypes they observe in culture have any physiological relevance or relevance to the hematopoietic phenotypes they observe in vivo.

Pt by pt response:

Reviewers' comments:

Reviewer #1 (Remarks to the Author):

The revised manuscript deals with many of the comments raised on the initial reviews. The additional data and tempering of the strength of conclusion improve the manuscript.

We thank the reviewer for their positive assessment of our revised manuscript. Based on Reviewer 2's comments, we have further tempered our conclusions, which we hope improves the manuscript.

Specific comments:

line 156-158. The authors should reword the text as I do not believe this is the intended meaning of the comments that were part of the initial review - Not "bias from the Cre locus" - the control is needed for toxicity from Cre expression and activity (DNA damage etc) in addition to a control for the plpC treatment and IFN response. The data should be shown as a supplemental file as it is an important control for readers to assess.

We thank the reviewer for the comment. We have now removed the phrase "bias from the Cre locus" and added this comment to the manuscript. We also included the heterozygous data as Supplemental Figure 1S-U.

Regarding Hypertribe (in comments to reviewers) - whilst unreasonable to change or complete more experiments, the authors should consider a deaminase dead ADAR fusion rather than EV as a control. I am not sure comparison to an EV is particularly useful.

We thank the reviewer for this comment and appreciate for not asking for an additional experiment. While we agree that a deaminase dead ADAR fusion is a good control, data from the original method papers (McMahon et al, Cell 2016; Xu et al, RNA 2018) showed that the deaminase dead ADAR is completely abolished of its activity, resulting in minimum editing signals equivalent to the empty vector background. In our experiment, EV was also used to control for any effect of transducing and expressing a retro-viral backbone in RNA editing in HSCs and MPPs. More importantly, EV provides a background of endogenous editing, which we have previously found to be equivalent to the editing levels of expressing ADAR alone (Nguyen et al., 2020). Therefore, this provides a helpful control to identify significant targets.

CDC42 over-expression data (in comments to reviewers) - I would encourage that this data be included as supplemental data. It is very important to acknowledge the limitations and uncertainty of the findings as well as the clear results so this would be important to include as it suggests more complexity than implied by the direct model proposed in Figure 7 (I dont see this as a flaw - I think it is better to accept that there is grey zones in results and acknowledge these). Related to this figure the WT data is hard to see due to the thickness of the line on the columns being the same colour as the individual sample dots.

We thank the reviewer for appreciating the results and have now added this important data to supplement figure 6M. We included the text to provide the context as we agree that it is interesting but has some limitations and uncertainty.

New data in Supp Fig S1S-U - this conclusion could have been even stronger with a Mx-Cre+ Wt control.....

Re plpC and genomic deletion - the authors have made an attempt to answer this. The references provided are nearly all from the lab that generated the current study so the statement of "many groups" is open to interpretation. I can accept this point and move on.

We thank the reviewer for their understanding to allow us to let our points stand. The heterozygous does provide a control for floxed allele, plpC and Cre toxicity and we observe a modest phenotype using this control. There are several studies (Kim et al., 2015; Liang Fei et al., 2018; Luscher-Frizlaff et al., 2019; Miyagi et al., 2019) that have published the phenotype of the WT Cre plpC control.

Kim, E., Ilagan, J.O., Liang, Y., Daubner, G.M., Lee, S.C.-W., Ramakrishnan, A., Li, Y., Rock Chung, Y., Micol, J.B., Murphy, M.E., *et al.* (2015). SRSF2 Mutations Contribute to Myelodysplasia by Mutant-Specific Effects on Exon Recognition. *Cancer Cell* 27, 617-630.

Liang Fei, D., Zhen, T., Durham, B., Ferrarone, J., T., Z., Garrett, L., Yoshimi, A., Abdel-Wahab, O., Bradley, R.K., Liu, P., *et al.* (2018). Impaired hematopoiesis and leukemia development in mice with a conditional knock-in allele of a mutant splicing factor gene U2af1. *PNAS* 115, E10437-E10446.

Luscher-Frizlaff, J., Chatain, N., Kuo, C.-C., Braunschweig, T., Bochynska, A., Ullius, A., Denecke, B., Costa, I.G., Koschmieder, S., and Luscher, B. (2019). Hematopoietic stem and progenitor cell proliferation and differentiation requires the trithorax protein Ash2l. *Scientific Reports*.

Miyagi, S., Sroczynska, P., Kato, Y., Nakajima-Takagi, Y., Oshima, M., Rizq, O., Takayama, N., Saraya, A., Mizuno, S., Sugiyama, F., *et al.* (2019). The chromatin-binding protein Phf6 restricts the self-renewal of hematopoietic stem cells. *Blood*, 2495-2506.

However, the data that is shown (Fig 6A-B) requires clarification - how am I meant to interpret this data. Whats the baseline and what is null in terms of signal? If I am assuming no protein is a score of 0 (this may be naive but the data is open to this interpretation) then the average KO cell has ~50% protein signal of a control cell (protein intensity of 1)?? I do not find this a clear answer to the question but it may be I am not interpreting it as the authors intended. The immunoblots are more easily understood.

We thank the reviewer for bringing to our attention this point. The reviewer is correct that there is a ~50% reduction of protein signal in the KO cell IF and has interpreted it correctly. The reason for why there is this apparent discrepancy between the assays is that the SYNCRIP antibody recognizes both endogenous SYNCRIP and HNRNP-R (as pointed out in immunoblot results). We used the same antibody for IF. This resulted in background signal coming from staining of HNRNP-R since the antibody detects both RNA binding proteins. We have now added this point to help the reader understand why residual signal remains in the IF.

Reviewer #2 (Remarks to the Author):

The authors have improved their paper by softening their conclusions; however, the data remain over-interpreted in multiple ways.

We thank the reviewer that stated that we have improved our paper. To address the remaining issue, we have reexamined our interpretations, revised the text to provide additional discussion, and included the perspectives that were raised by the reviewer. On this point, we have further edited the title to **“RNA binding protein SYNCRIP maintains proteostasis and self-renewal of hematopoietic stem cells and progenitor cells.”** Also, we have tempered several of our main points. Most importantly, we have reworked the focus of the manuscript to emphasize all of the defects of the SYNCRIP mice, deemphasize the reserve HSC concepts towards a defect in self-renewal in both HSCs and progenitors. We believe that this new focus has strengthened our manuscript and provided a more balanced perspective on our data. We thank the reviewer for their perspective.

1. One of the key issues during the last round of review was that the authors were arguing that SYNCRIP was preferentially required by “reserve” hematopoietic stem cells (HSCs) even though the data showed that SYNCRIP is ubiquitously expressed by hematopoietic cells and functionally necessary in many different stem and progenitor cell populations, such that its deletion broadly reduces blood cell counts, even under steady state conditions. In the current manuscript they have softened this claim, but it is still made. For example the abstract starts “Tissue homeostasis is maintained after stress by engaging a subset of dormant and highly self-renewing stem cell populations”. In reality, stress activates all HSCs, not just a subset of dormant stem cells.

We agree with the reviewer that all HSCs and not just a subset of HSCs respond to stress. Thus, we have now edited the manuscript further to clear up this confusion in our description. We revised the phrase to “Tissue homeostasis is maintained after stress by engaging and activating the hematopoietic stem and progenitor compartments in the blood”.

They go on to identify a previously uncharacterized “atypical HSC” population that they concluded is “specifically affected” by SYNCRIP deletion. However, there is no functional evidence that the “aHSCs” that were identified based on gene expression profiling are actually HSCs. They could be MPPs or early restricted progenitors, which are also affected by SYNCRIP deficiency.

We respectfully disagree with this comment as the aHSC population was defined based on gene expression profiling to correspond to HSCs, and in particular to the more dormant/reserve low-input HSCs (Rodriguez-Fraticelli et al., 2020) more closely. The low-input HSCs characterized by in Rodriguez-Fraticelli et al., 2020 was in fact demonstrated functionally to be the population responsible for reconstitution in secondary transplantation. Based on our extensive gene expression analysis, the aHSCs are not enriched to be MPPs by any measure and MPPs do not have long-term self-renewal and serial transplantation capacity. Moreover, our functional in vitro assays and in vivo transplant assay demonstrate a role for SYNCRIP’s function in the entire HSC compartment. We have edited the manuscript to temper comments about specific effects in aHSCs as we agree that defects were observed across the HSC compartment.

It is not clear what these aHSCs are and the data definitely don’t support the idea that they are specifically affected by SYNCRIP deficiency. Thus, the authors continue to ignore the obvious conclusion that SYNCRIP is broadly required by stem and progenitor cells in favor of the flashier idea that there is a novel subset of dormant HSCs that is preferentially

affected. The authors have neither shown that the atypical HSCs are actually HSCs, nor that they are preferentially affected by SYNCRIP deficiency.

We apologize that the reviewer believes that we have ignored the finding that there is a broad defect in stem and progenitor cells. While we already included the description of HSCs and progenitors' phenotypes in our manuscript, we agree that it might not be enough. We have now further revised some of the wording to deemphasize that there is a specific defect within only the aHSCs. We have also revised the results and discussion to highlight these important points and broaden the description of the defects observed in other compartments.

However, as addressed in the previous comment, our data supports that atypical HSCs are clearly HSCs (more closely associated with long-term and reserve low-input HSCs) based on their gene expression. Moreover, dormant/reserve HSCs are quiescent and by their definition able to serially transplant and engraft with multilineage capacity. Based on these criteria the SYNCRIP HSCs are defective regardless of the gene expression assessment of the aHSC population. The defect at the level of the HSC was demonstrated through long-term engraftment assessment as well as secondary transplant studies and examining the HSC compartment in these mice. We in fact concluded that "the phenomenon could be specific for HSC and aHSC clusters", indicating that this is an HSC phenotype on top of a progenitor defect.

2. The authors reasoning that they "inferred that the activity loss observed in the secondary engraftment was largely driven by the "reserve" HSC compartment is not sound. When defects in stem/progenitor cells lead to reduced blood cell counts, there is a broad activation of HSCs and hematopoiesis in an effort to restore normal blood cell counts. This broadly depletes the self-renewal potential of HSCs, reducing their reconstituting potential, particularly upon serial transplantation. This phenotype has been observed in scores of mutant mice and never shown to be driven only by effects on "reserve" HSCs.

We thank the reviewer for their critical assessment and agree that repeated and increased HSC activation results in the loss of self-renewal. However, it is important to note that SYNCRIP's phenotype does not show HSCs to be more activated after SYNCRIP loss. We conclusively demonstrated that 1) The HSCs are not proliferating more nor is there a difference in the frequency of cells that have exited quiescence. 2) HSC defects are cell intrinsic and demonstrate a clear activated stress response and not a proliferative program: a) gene expression and i.e stress signatures, b) reduced in vitro self-renewal and replating activity even in primary animals with almost no changes in steady state, c) in vitro assays indicating modest changes in proliferation but altered fate decisions, d) competitive transplants demonstrating HSC loss in the context of normal host cells. Therefore, while the interpretation the reviewer raised on loss of self-renewal due to broad activation of HSCs is possible, it does not apply for the phenotypes observed in SYNCRIP deficient animals. However, to provide a counter-balanced discussion of our data and mechanisms, we have revised the text to include this point made by the reviewer that there is a formal possibility of the defect in self-renewal partially driven by more activation of progenitors. Moreover, we made it clearer that there is a self-renewal defect in SYNCRIP HSC and progenitor compartment. We hope the revision satisfies the reviewer's perspective.

3. The authors have also softened their claims about CDC42, acknowledging now that it

gives only a partial rescue; however, I remain uncomfortable with their claims about effects of SYNCRIP on “cell polarity” and asymmetric division for the reasons described in my original review. It is not clear whether the phenotypes they observe in culture have any physiological relevance or relevance to the hematopoietic phenotypes they observe in vivo.

Based on the reviewer 1’s comment, we have included our partial rescue into the supplement figure 6M and revised the text to provide further context and tempering our conclusions. However, it is important to note that the *in vitro* assays demonstrate intrinsic defects in the HSC compartment and thus provide an exciting cellular mechanism for our *in vivo* findings. Additionally, the loss of self-renewal in stem progenitor replating is shared with our *in vivo* self-renewal defects. It is an important challenge in the field that direct characterization of asymmetric division and tracking of cellular polarity remain not feasible *in vivo*. However, as we pointed out in our previous response to the reviewer, while the evaluation of symmetric vs. asymmetric division was performed in an *in vitro* setting, functional assessment of HSCs in the daughter assays was examined by single cell micromanipulation and single cell transplantation into recipient animals. In several studies (Ito K et al.2012, Yamamoto R et al. 2013, Ito K et al. 2021), two daughter cells in the paired daughter cell assay were isolated, transplanted into mice and followed for their ability to reconstitute in primary and secondary recipients. Isolated cells were then retrospectively defined as long-term, intermediate-term or short-term HSCs. The evidence presented by these studies strongly supported the functional relevance *in vivo* of HSC polarity and asymmetric cell division. Thus, while we are limited to make some of these correlations based on *in vitro* data, the literature supports our conclusion that the observed altered polarity and cell division are relevant with our observed *in vivo* data. At the same time, we have added text to the discussion to make the reviewer’s point on the need for further examination *in vivo* of altered polarity to definitively support the conclusions.

REVIEWER COMMENTS

Reviewer #3 (Remarks to the Author):

The authors identified a potential regulator of the depth of HSC quiescence. They have performed huge amount of work. However, they missed the opportunity to fully expose their finding on HSC quiescence. It is not totally convincing that Syncrip deleted mice do not have an HSC phenotype, under homeostasis. Specifically, there is an increase in the LSK compartment that the authors do not discuss. Other elements include the BM cellularity, RBCs ad HGB, and the frequency of HSCs in G0 (in deleted mice, HSCs seem to be found more in G1). By increasing the n of the mice the authors might see significant, albeit mild, differences between groups. This would give more confidence in Syncrip as only required during stress.

The depth of HSC quiescence may impact the response to stress, depending on the type and stress levels. The authors should take advantage of published findings (PMID: 26674251; PMID: 27731316) to deepen their results on the role of Syncrip in HSC quiescence.

The functional characterization of "atypical HSCs" is missing. Significant gene expression similarity does not support functional identity. It will be important to limit their claim of "atypical HSCs" only to their genomic analyses.

As a side note, coining the HSCs population they identified as atypical HSCs or "aHSCs" is very confusing, not a good idea, given that aHSCs are known in the field as active HSCs, a label used by Andreas Trumpp (Cabezas-Wallschild, 2017) and others and the opposite of the authors claim.

The authors should include in their references PMID: 19062086; PMID: 24749072

Reviewer #3 (Remarks to the Author):

The authors identified a potential regulator of the depth of HSC quiescence. They have performed huge amount of work.

We appreciate the reviewer for their positive feedback.

However, they missed the opportunity to fully expose their finding on HSC quiescence. It is not totally convincing that Syncrip deleted mice do not have an HSC phenotype, under homeostasis. Specifically, there is an increase in the LSK compartment that the authors do not discuss.

We thank the reviewer for this comment. We would like to clarify that we observed no severe effects in HSC frequency or absolute numbers of HSCs and multipotent progenitors (MPPs) upon SYNCRIP depletion, but we did observe an increase in LSK frequency under steady state conditions. We have modified the wording and included additional discussion in the manuscript to make this clear. On the other hand, while there was not a significant difference in HSC frequency at homeostasis, their ability to repopulate was modestly reduced.

Taken the suggestion of the reviewer, to further assess HSC phenotypes at steady state, we conducted cell cycle analysis using $n=5$ for each group of Syncrip WT f/f vs. Syncrip KO Δ/Δ . We found no significant change in frequency of HSCs at G0, G1 or S/G2/M phase of the cell cycle while we observed population we see a slight increase in G1 phase with $p=0.068$ within MPP1 (data shown below). While we agree with the reviewer 3 that there might be a trend of increase in G1 and S/G2/M, the data however pointed to a rather subtle change that was not consistent in our animal cohort, suggesting if there is any, effects of SYNCRIP loss on cell cycle and quiescence at the homeostatic state is likely modest.

Other elements include the BM cellularity, RBCs ad HGB, and the frequency of HSCs in G0 (in deleted mice, HSCs seem to be found more in G1). By increasing the n of the mice the authors might see significant, albeit mild, differences between groups. This would give more confidence in Syncrip as only required during stress.

We agree with the reviewer's point that adding more repeats will increase the power of our statistical test. A dataset comprising of $n=5$ showed a small but significant increase in LSK frequency (Figure 1E- 3-week time point) but no significant difference in HSC frequency (Figure

1F). Similarly, we observed significant decrease in total WBC and lymphocyte count but not other compartments (supplemental figure 1C-G). In our cell cycle flow analysis using PY and Hoestch staining, we included n=6 (individual mice) for each group and in fact were able to demonstrated statistically significant difference in cell cycle of MPP population while the n=6 did not reach significant within the HSC compartment. In addition, we performed an orthogonal method using CSFE to label cycling cells in vivo and included n=5 mice for each group. Using this method, we also observed no significant change in cell cycle activity of HSCs. Taken together, we assayed for cell cycle activity of HSCs in total 11 mice of each genotype. The statistical power of the cohort is sufficient to uncover the most significant phenotypes upon SYNCRIP knockout. Additionally, many assessments were performed at 24 weeks post-plpC, which precludes our ability to conduct the experiments in the timely manner. To make the point clearly, we included in our manuscript the statement that **“SYNCRIP has modest effects in steady state hematopoiesis”**.

The depth of HSC quiescence may impact the response to stress, depending on the type and stress levels. The authors should take advantage of published findings (PMID: 26674251; PMID: 27731316) to deepen their results on the role of Syncrip in HSC quiescence.

We thank the reviewer for the suggestion and have added these references and interesting studies. We agree with the reviewer that the depth of HSC quiescence may impact the response to stress. These studies focus on exploring the connection between mitochondrial metabolism and HSC self-renewal. Specifically identifying that low mitochondrial activity or mitochondrial membrane potential promotes HSC self-renewal. However, our data pointed to a minor impact of SYNCRIP on HSC's quiescence as demonstrated by cell cycle analysis, BrdU incorporation and CFSE labeling. Moreover, while we observed a strong induction of the UPR and HSF pathways in HSC upon SYNCRIP depletion, no significant change at the transcriptional level was observed in pathways related to mitochondrial activity in our single cell RNA-seq. Interestingly, we found multiple mitochondrial and respiratory electron transport related pathways in our single cell RNA-seq analysis of MPPs, suggesting MPPs may have increased mitochondrial activity upon SYNCRIP depletion. In addition, we do observe upregulation of respiratory electron transport pathways in Syncrip depleted HSCs vs. control in the bulk HSC RNA-seq. However, we did not find a significant enrichment for genes in these pathways as direct SYNCRIP targets based on the HyperTribe mapping of SYNCRIP's binding transcripts. Taken together, the data suggested that the phenotypes observed with mitochondrial metabolism are likely a downstream and indirect effect of SYNCRIP loss of function. We have now included this in the discussion and highlighted the observation among the remaining future questions from this study.

The functional characterization of “atypical HSCs” is missing. Significant gene expression similarity does not support functional identity. It will be important to limit their claim of “atypical HSCs” only to their genomic analyses.

We thank the reviewer for their feedback on this portion of our study. We agree that without functional assessment of this HSC population it is difficult to make many claims, and therefore we have further modified the manuscript to put less emphasis on this population. Also, important to note that our downstream assessments on protein quality, HSC cell polarity and cell division was done on phenotypic HSCs (LSK, CD150+ CD48-) and not specifically on this population therefore SYNCRIP's effects are not specific to only one population of HSCs. We have now tempered this point and have explicitly stated that this cluster is limited to our gene expression and no functional assessment of this population was performed.

As a side note, coining the HSCs population they identified as atypical HSCs or “aHSCs” is very confusing, not a good idea, given that aHSCs are known in the field as active HSCs, a label used by Andreas Trumpp (Cabezas-Wallschield, 2017) and others and the opposite of the authors claim.

We appreciate the reviewer pointing this out to us and we agree, this could cause some confusion. We have therefore re-named these populations HSC-cluster 1 (HSC-C1) and HSC2-cluster 2 (HSC-C2) so they are not associated with the previously identified active HSC (aHSC) and further suggest that this population is part of our genomic analysis and not functionally assessed.

The authors should include in their references PMID: 19062086; PMID: 24749072

These studies have now been referenced in the manuscript.